**Technology**

# Hist2Cell: Deciphering fine-grained cellular architectures from histology images

## Graphical abstract

## Authors

Weiqin Zhao, Zhuo Liang, Xianjie Huang, Yuanhua Huang, Lequan Yu

## Correspondence

yuanhua@hku.hk (Y.H.),
lqyu@hku.hk (L.Y.)

## In brief

Zhao et al. present Hist2Cell, a vision graph-transformer framework to decipher fine-grained transcriptional cellular architectures directly from histology images. Hist2Cell enables scalable, super-resolved cellular mapping and precise cancer prognosis by capturing complex cell-cell colocalizations, offering a cost-efficient tool for large-scale spatial biology studies on routine clinical slides.

## Highlights

- Hist2Cell deciphers fine-grained transcriptional cell types from H&E images

- Hist2Cell generalizes robustly to external datasets and large-scale cohorts

- Hist2Cell enables super-resolved analysis and precise cancer prognosis

Zhao et al., 2026, Cell Genomics 6, 101137
March 11, 2026 © 2026 The Authors. Published by Elsevier Inc.

# Cell Genomics

CellPress

## Technology

# Hist2Cell: Deciphering fine-grained cellular architectures from histology images

Weiqin Zhao,[1] Zhuo Liang,[1] Xianjie Huang,[2,3] Yuanhua Huang,[1,2,3,*] and Lequan Yu[1,4,*]

[1]School of Computing and Data Science, the University of Hong Kong, Hong Kong SAR, China
[2]School of Biomedical Sciences, the University of Hong Kong, Hong Kong SAR, China
[3]InnoHK-Centre for Translational Stem Cell Biology, Hong Kong Science Park, Hong Kong SAR, China
[4]Lead contact
*Correspondence: yuanhua@hku.hk (Y.H.), lqyu@hku.hk (L.Y.)

## SUMMARY

Histology images offer a cost-effective approach to predicting cellular phenotypes using spatial transcriptomics. However, existing methods struggle with individual gene expression accuracy and lack the capability to predict fine-grained transcriptional cell types. We present Hist2Cell, a vision graph-transformer framework to accurately resolve fine-grained cell types directly from histology images. Trained on human lung and breast cancer datasets, Hist2Cell predicts cell-type abundance with high accuracy (Pearson correlation over 0.80) and captures cellular colocalization. Moreover, it generalizes to large-scale The Cancer Genome Atlas (TCGA) cohorts without re-training, facilitating survival prediction by revealing distinct tissue microenvironments and cell type-patient mortality relationships. Thus, Hist2Cell enables cost-efficient analysis for large-scale spatial biology studies and precise cancer prognosis.

## INTRODUCTION

The cellular architecture of tissues, where distinct cell types are organized in space, underlies cell-cell communication, organ function, and pathology, enabling our broad and deep understanding of diverse cellular behaviors and biology. For example, cell-cell communication plays an essential role in various biological processes and functional regulations,[1,2] such as immune cooperation in a tumor microenvironment, organ development and stem cell niche maintenance, and wound healing. Although previous digital pathology-based methods[3–5] could segment and identify different cells, they can only identify coarse-grained cell-type groups (usually 2–4 major cell types) and provide limited information for detailed analysis. Fortunately, emerging spatial transcriptomics (ST) technologies provide key opportunities to map fine-grained transcriptional (for instance, 80 cell types in lung tissue[6]) resident cell types and cell signaling *in situ* in a scalable manner by probing the full transcriptome, for example, Slide-seqV2[7] and Stereo-seq.[8] Recently, several computational methods have been proposed for analyzing ST data to reveal cell variability in a tissue context. The most critical step is to characterize the fine-grained transcriptional cell type proportion (or abundance) in each ST spot, and one of the most popular strategies is to integrate ST with the reference transcriptome signatures of cell types obtained from relevant single-cell RNA sequencing (scRNA-seq) profiles. These methods started from general data interaction methods, such as Seurat v.3,[9] but soon embraced tailored designs for ST by modeling the read counts with negative binomial distributions (stereoscope,[10] RCTD,[11] and Cell2location[12]), or via non-negative ma-

trix factorization (spotlight[13]), or as a mapping from individual cells (e.g., Tangram[14] and TransImp[15]). Of note, among these methods, Cell2location uniquely provides an estimate of the absolute cell abundance on top of its proportion; also, its Bayesian treatment enables a higher sensitivity and resolution by accounting for uncertainty in technical sources of variation and borrowing statistical strength across locations; therefore, it was employed for cell-type abundance estimation here.

However, current ST technology cannot be scaled due to high cost, while existing methods that aim to predict directly from histology images have generally been reported with limited accuracy. To address these challenges, we present Hist2Cell, a graph-transformer framework to resolve fine-grained transcriptional (up to 80) cell types directly from hematoxylin and eosin (H&E)-stained histology images and create cellular maps of diverse tissues with a one-stage pipeline. Hist2Cell was trained on spatial transcriptome datasets for human lung[6] and human breast[16] cancer with spot-level cell abundance estimations from the Cell2location[12] algorithm. For held-out test patients, Hist2Cell can accurately predict the spatial variations in the localization of 40 fine-grained transcriptional cell types—at a resolution of around 100 μm, serving as important references for biological research. Specifically, it can predict substantial variation in cell localization within regions labeled as tumors or different tissue structures by clinicians, demonstrating that Hist2Cell captures significant tissue heterogeneity. By feeding the predictions into an existing toolbox such as SpatialDM,[17] Hist2Cell can also reveal accurate cell-type colocalizations, prioritizing communicating cells and identifying interaction spots in the spatial context. Hist2Cell is more accurate than deep

learning methods that predict cell abundance solely from a single histology image spot or a limited neighboring context, as it leverages both local and global correlations of histology image spots with the proposed graph-transformer architecture while maintaining data diversity with a random subgraph sampling strategy.

As an independent external test, Hist2Cell accurately predicts the spatial distribution of 15 fine-grained transcriptional cell types and their localization in breast cancer data from an external source[18] without any framework modification or fine-tuning. This suggests that Hist2Cell can robustly deal with the unseen samples despite potential batch effects or technique differences in imaging. As a result, Hist2Cell can provide consensus cellular analysis for biologists on the diverse large cohort studies (e.g., The Cancer Genome Atlas [TCGA] cohorts) that current ST technology cannot scale due to high cost. More importantly, Hist2Cell's fine-grained transcriptional cell abundance predictions could support precise cancer prognosis. In this study, we demonstrate that it outperforms the previous state-of-the-art pathology models in survival risk prediction tasks on three common cancers and presents promising potential to serve as a cheap yet comparable alternative to expensive bulk RNA-seq for cancer prognosis in real clinical settings. Moreover, leveraging integrated gradients, Hist2Cell uncovers the relationships between different cell populations and patient mortality, which could contribute to validating existing cancer research and provide new biological insights. In addition, Hist2Cell could provide the above analysis/application under a higher spatial resolution (e.g., 16×, near the single-cell level) than the original spatial transcriptome technique by utilizing low-resolution cellular maps or directly predicting from histology images. In general, Hist2Cell paves the way for reliable investigations of cellular organization and communication in diverse tissues and for precise survival prediction of cancer prognosis in a cost-efficient manner, with potential broader applications in diagnostics and personalized medicine.

## DESIGN

### Enhancing ST analysis with histology images
Considering that current sequencing-based ST technologies generally suffer from low spatial resolution and low RNA capture efficiency (and sequencing depth), recent efforts have also been made to take advantage of the paired histology image to enhance ST analysis. Prominent examples include XFuse, which uses a variational autoencoder to generate a super-resolution transcriptome by sharing a latent variable between ST and its image,[19] while TESLA[20] and iSTAR[21] further accelerate the computational efficiency of this task via graph-based smoothing. Additionally, SpatialScope[22] utilizes histology to segment cell nuclei and achieves single-cell-level transcriptome resolution through a deep generative model (i.e., diffusion model).

However, a key limitation of these existing imputation-like approaches is the requirement of estimating from spatial transcriptome data from the same or consecutive tissue sections of the same patient, which hinders large-scale studies due to the high costs. Relatively, histology images are much cheaper and easier to acquire and are routinely used in clinics.[23] Previous studies have shown that histological images are highly corre-

lated with gene expression and cell-type identification, and they have been used to predict bulk gene expression profiles,[18] coarse-grained cell-type abundance,[3] and genomic aberrations.[24] Furthermore, powered by ST technologies that provide a large amount of high-quality training data, multiple attempts were made to directly predict the expression of a set of critical genes or the full transcriptome from a cell-sized path of images.[18,25–28] However, these methods, which aim to predict the expression of a small group of individual genes, were generally reported with limited accuracy, probably due to the lack of predictive information for most genes. Therefore, their predictions are unlikely to serve as reliable sources for estimating fine-grained transcriptional cell abundances, and how to identify fine-grained transcriptional cell-type spatial variations and infer cell-cell communication directly from histology images is under-explored and remains an open problem. Moreover, existing methods are also incapable of creating more detailed cellular wiring diagrams for higher-resolution analysis when estimating from spatial transcriptome data, as they ignore the information provided by the histology images.

### Hist2Cell: A one-stage framework for fine-grained cell-type prediction
Hist2Cell is constructed to address these limitations by introducing a one-stage predictive framework. Instead of performing a reference-dependent estimation for every new sample, we reframe the problem into a supervised learning task. We leverage highly accurate cell abundance estimations from Cell2location, not as an inference step but as a one-time operation to generate high-quality ground-truth labels for model training. Once trained, Hist2Cell operates as a standalone predictive model that resolves fine-grained transcriptional cell types directly from H&E images, decoupling the prediction task from the need for concurrent molecular data.

To execute this direct prediction, we considered that current approaches can be divided into two groups: (1) global-based methods and (2) local-based methods. Global-based methods suffer from severe data scarcity and high computational cost, while local-based methods do not take into consideration characteristics such as the vicinity of the patch or long-range interactions, resulting in sub-optimal performance.

Building on these findings, Hist2Cell employs a graph-transformer architecture to learn both local context and distant relations, thus benefiting from the advantages of both approaches without succumbing to their respective limitations. The key idea is to execute a minibatch of local subgraphs sampled from a whole-slide image (WSI) instead of using an entire WSI as a single data element. To better model the relationships, the graph attention layers facilitate message passing between spots and their spatial neighbors, learning the local spatial context, while the transformer layers model long-range correlations. This allows Hist2Cell to be more accurate than methods that predict cell abundance solely from a single histology image spot or a limited neighboring context, as it leverages both local and global correlations of histology image spots with a random subgraph sampling strategy.

To avoid confusion with the abovementioned prior work, a detailed comparison among Hist2Cell and previous works is

**Table 1. Conceptual comparison among methods**

| Features | Group/method | | | | |
| --- | --- | --- | --- | --- | --- |
| | 1 | 2 | | 3 | Ours |
| | Classic DPath | HER2RNA | ST prediction | XFuse, iStar | Hist2Cell |
| **Input** | | | | | |
| Histology image | ✔ | ✔ | ✔ | ✔ | ✔ |
| Spatial gene | × | × | × | ✔ | × |
| **Label source** | | | | | |
| Manual annotation | × | × | × | × | × |
| Spatial gene | × | × | ✔ | ✔ | ✔ |
| **Output** | | | | | |
| Cell abundance | ✔ | N/A | N/A | N/A | ✔ |
| Cell granularity | 2–4 types | N/A | N/A | N/A | up to 80 types |
| **Generalize** | | | | | |
| Within section | ✔ | ✔ | ✔ | ✔ | ✔ |
| Cross-patient | ✔ | ✔ | ✔ | × | ✔ |

Conceptual comparison between our Hist2Cell with classic digital pathology (DPath), HE2RNA 779/ST prediction works, and XFuse/iStar works.

provided in Table 1. Briefly, Hist2Cell is the first method that leverages ST as training to infer fine-grained transcriptional (up to 80) cell types as more predictable phenotypes in new tissue/patient samples. All existing methods either predict only 2–4 cell types (group 1: conventional digital pathology), focus on noisy gene expression (group 2: ST-Net and variants/HE2RNA), or impute the training samples (group 3: iStar/XFuse). The conceptual design and architectural rationale that enable these capabilities are detailed in the following section.

## RESULTS

A schematic figure illustrating the workflow of Hist2Cell is presented in Figure 1A, along with a summary of the example downstream tasks depicted. For the detailed operation, as shown in Figure 1B, initially, Hist2Cell crops the entire slide image into a multitude of spot patches, with a spot size of $150 \times 150 \ \mu m^2$ according to the ST platforms. Treating spots as nodes and their spatial neighboring relationships as edges, each whole-slide image is formulated into an independent graph. Subsequently, during the iteration of constructed graphs, multiple subgraphs are randomly sampled from a certain whole-slide image to serve as training inputs for Hist2Cell. Each subgraph consists of a central node and its k-hop spatial neighborhood. Following this, an image encoder extracts node embeddings from the images constituting each subgraph. Hist2Cell then employs a graph-transformer architecture to learn and integrate both short-range local and long-range global relationships among a batch of subgraphs from the same whole-slide image, thereby enhancing the prediction of cell abundance for each node. The image encoder and the graph-transformer architecture are concurrently trained in an end-to-end fashion, minimizing the distance between the cell abundance predictions and ground-truth labels estimated from the spatial transcriptome data. In the inference phase, Hist2Cell uses the predictions of the center nodes as the conclusive prediction and implements a sliding-window strategy to ascertain the cell abundance prediction for all spots. To assess

the effectiveness of Hist2Cell, we study healthy human lung and human breast cancer tissues and large-scale TCGA histology image datasets.

## Hist2Cell maps cellular architectures and detects key colocalized cell types in human lung tissues

To validate Hist2Cell, we initially applied the framework to data from 5 proximal-to-distal locations in healthy human lungs from 4 donors.[6] This dataset provides 11 H&E-stained histology slides, resulting in a total of 20,770 spots with 80 fine-grained transcriptional cell-type abundances estimated by the Cell2location algorithm according to the authors. More details about the 80 cell types can be found in Figure S1. All performance comparisons, visualizations, and analyses are reported using leave-one-donor-out cross-validation, where we iteratively trained Hist2Cell on three of the donors and made predictions on the remaining held-out donor.

We applied Hist2Cell for cell-type identification. As depicted in Figure 2A, the performance of Hist2Cell was compared with methods designed for predicting local gene expression from histology images, including STNet,[18] DeepSpaCE,[29] Hist2ST,[25] and THItoGene.[26] For these ST-prediction-based methods, we follow their default two-stage setting of first predicting spatial gene expressions and then using these predictions to estimate cell abundance. As shown in the figure, Hist2Cell demonstrated more accurate predictions of cell-type abundances across locations compared to other methods, as evidenced by the overall Pearson correlations of all folds. Specifically, the analysis revealed that for 79 out of 80 cell types, the predicted cell abundance from Hist2Cell positively correlated with the provided cell abundance labels across all patients, with an average Pearson's correlation of 0.31 among all 80 cell types and patients, demonstrating a performance increase in Pearson correlation coefficient (PCC) of approximately 50% improvement compared with the best baseline methods. To provide a more granular comparison, we generated histograms of the performance across all 80 cell types (Figures S2 and S3). This detailed analysis

**CellPress**

**Cell Genomics**
Technology

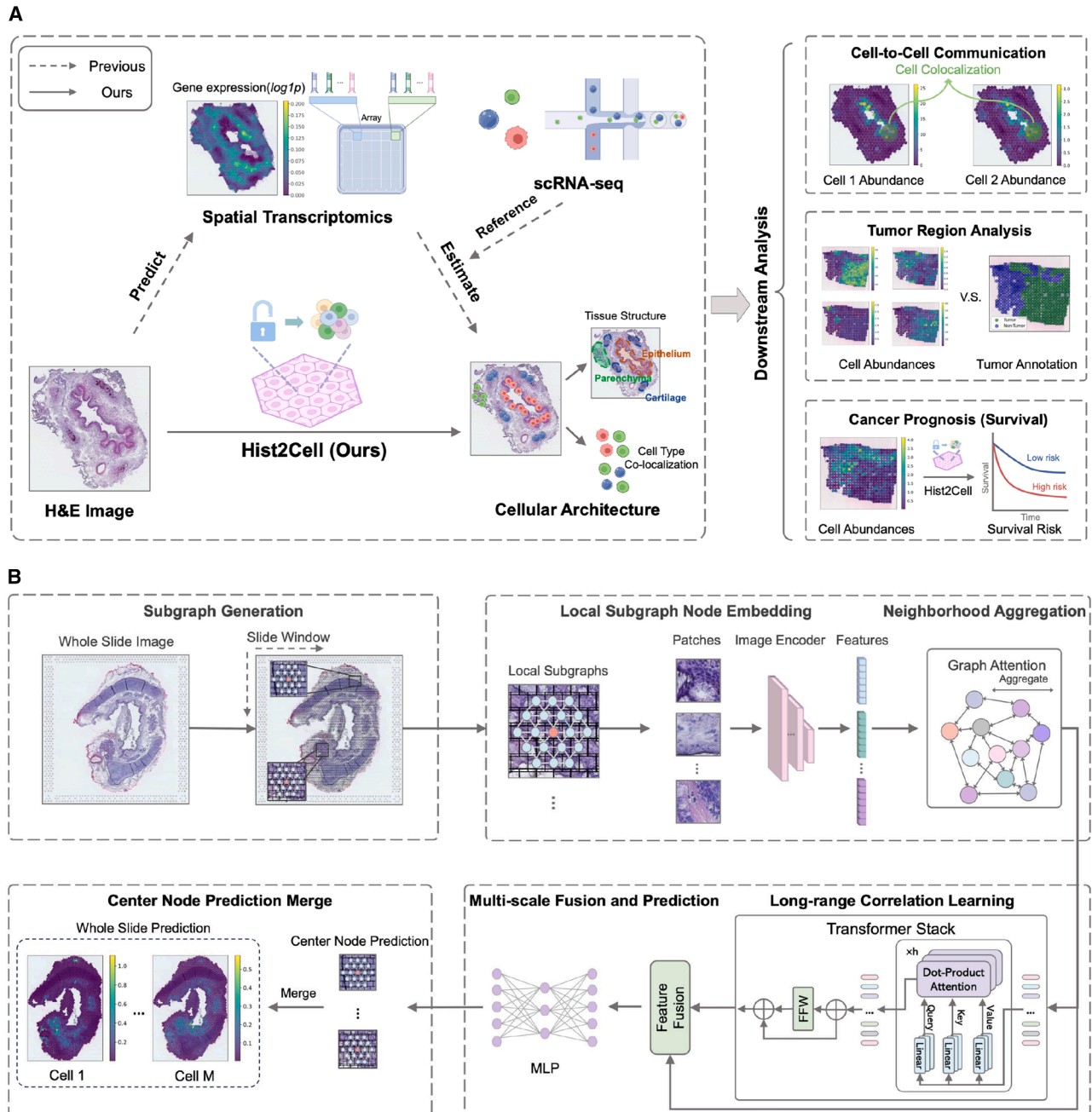

**Figure 1. Overview of Hist2Cell**

(A) Left: an illustration demonstrating how Hist2Cell directly predicts spatial cell abundance from histology images (one-stage), in contrast to how previous works estimate spatial cell abundance from spatial transcriptomics data using scRNA-seq data as a reference (two stage). Right: the three major downstream tasks based on spatial cell abundance predictions by Hist2Cell—encompassing cell-cell colocalization analysis, tumor region analysis, and cancer prognosis (survival risk prediction).

(B) Hist2Cell initially formulates the whole-slide image into a spatial graph, followed by sampling local subgraphs as input for the model. It employs a feature encoder, graph attention layers, and transformer layers to learn and extract the morphological features of the tissue, as well as to discern both local and long correlations among histology spots. Multi-scale features from these components are fused for fine-grained spot cell abundance prediction. The prediction for the central node of the sampled subgraphs is used as the final output during the inference phase.

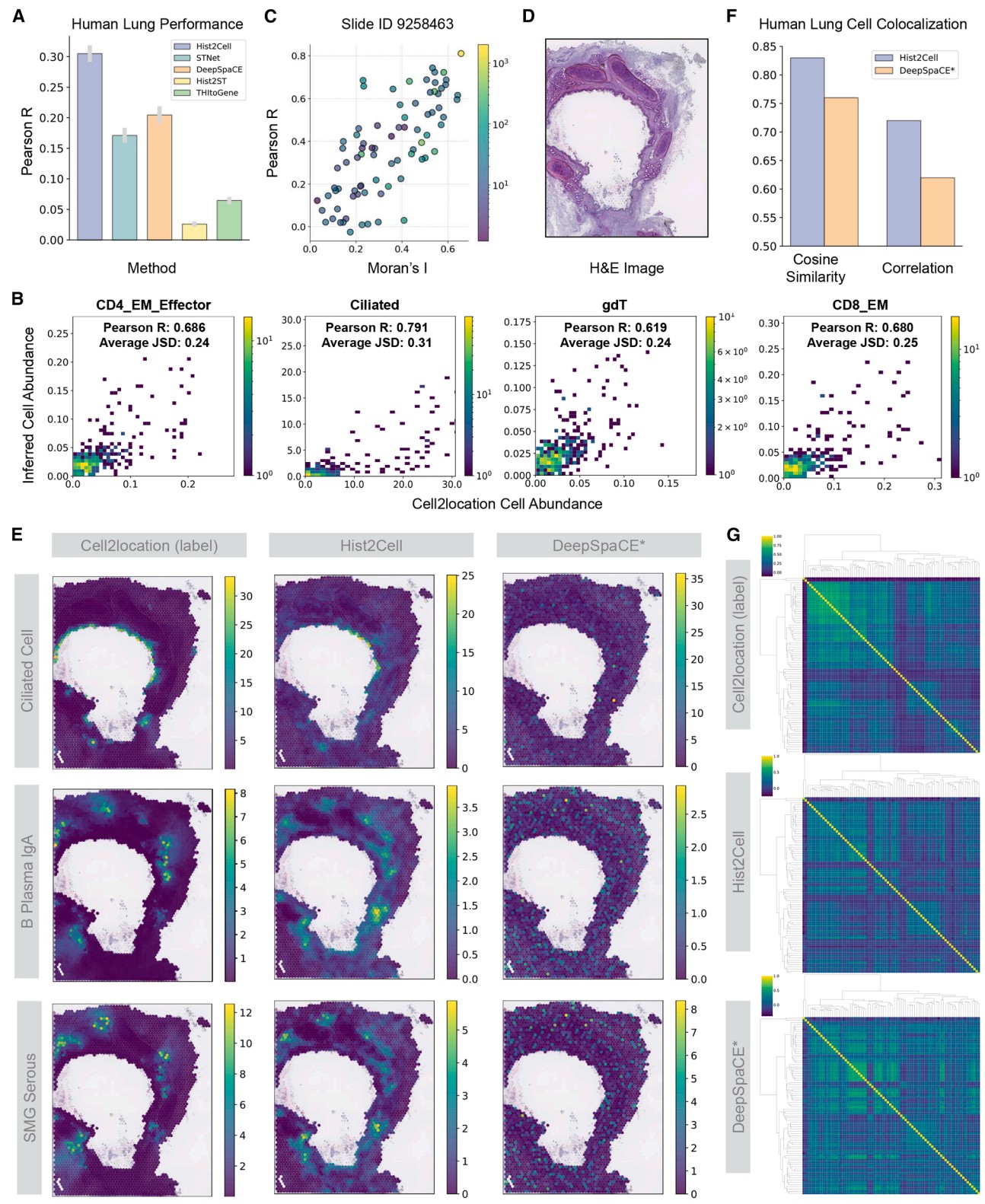

reveals that Hist2Cell outperforms the original baseline methods on the vast majority of cell types (73 out of 80). We also examined the top 30% cell types across 11 slides, and the average Pearson's correlation reached 0.50 vs. 0.43 by the best counterpart, as shown in Figures S4 and S5. Moreover, despite the divergence of different cross-validated slides, we identified a group of 40 fine-grained transcriptional cell types that can be predicted with high to moderate accuracy (a Pearson's correlation above 0.25 for at least 50% slides) by Hist2Cell. These results underscore the superiority of our methodology and the one-stage prediction strategy advantage, suggesting that Hist2Cell can directly decode fine-grained cellular architectures from histology images by modeling the tissue morphology feature, local context interactions, and long-range correlations with the slide. Specifically, we also adopt the proposed one-stage prediction strategy of Hist2Cell and only adopt the network architecture of the baseline methods as the compared counterpart; the results are shown in Figure S6. The adapted baseline methods gained instant performance increase, illustrating the efficacy of avoiding the noise contained in the ST predictions. Furthermore, our approach also surpasses "adapted" methods, highlighting the model advantage of our graph-transformer architecture and subgraph-sampling-based training paradigm. Particularly, we highlight that Hist2Cell showed significantly high positive correlations with several key and informative cell types, including CD4[+] effector memory T cells (CD4-EM/effector) with a Pearson's R of 0.68 and Jensen-Shannon divergence of 0.24, ciliated cells with a Pearson's R of 0.79 and Jensen-Shannon divergence of 0.31, CD8 EM cells with a Pearson's R of 0.68 and Jensen-Shannon divergence of 0.25, and gamma-delta T (gdT) cells with a Pearson's R of 0.62 and Jensen-Shannon divergence of 0.24 (Figure 2B). As shown in Figures S7 and S8, these results represent a remarkable improvement even when compared against the strongest "adapted" baseline methods, with performance gains of +0.16 for CD4-EM/effector and +0.28 for gdT cells. Note that these highly predictable cell types have critical functions in the human airway system; for example, CD8 EM cells are essential in directly eliminating infected or malignant cells *in situ*,[30] and ciliated cells play a pivotal role in airway homeostasis by trapping and expelling microorganisms, mucus, and other debris.[31] Systematically, we also found that Hist2Cell excelled in predicting the abundance for cell types exhibiting a higher spatial autocorrelation denoted by Moran's *I* (i.e., spatially variable cell types, as detailed in Figure 2C). In other words, on top of the reasonably high performance across all cell types, Hist2Cell achieves even more accurate prediction for cell types that may have specific spatial patterns, which often relate to biological significance.[32–34] Beyond the quantitative assessment, visualization of the predicted cell-type abundances in tissue also shows that Hist2Cell accurately identifies spatial patterns of cell-type distributions, with fewer false positives compared to other methods, for example, in ciliated cells, IgA plasma cells, and submucosal gland (SMG) serous cells (Figures 2D and 2E).

We next evaluated Hist2Cell in recognizing cell-type colocalization patterns; specifically, we examined the pairwise colocalization among these 80 cell types by performing a spatial correlation with bi-variant Moran's R as implemented in SpatialDM.[17] Briefly, this statistic calculates the correlation between the abundances of two cell types while considering spatial autocorrelation. For a direct numerical comparison, we visualized the cosine similarity and correlation between the colocalization calculated from the predictions of different models and the ground-truth cell abundances, and Hist2Cell excelled in both metrics, as shown in Figure 2F. Overall, we also found that the patterns of colocalization among these 80 cell types are highly consistent between the quantified abundance from ST and predicted ones from histology by our Hist2Cell (Figure 2G). By contrast, discordance is evident if using the cell-type abundance predicted by the counterpart method DeepSpaCE* (the best-performing adapted baseline method in Figure S6; * represents adapted baselines using a one-stage prediction strategy). As a prominent example, we noticed that Hist2Cell identifies colocalization between IgA plasma cells and SMGs in human airways, which is one of the major discoveries in the original study.[6] Impressively, with the observed cell-type abundance from ST, IgA plasma was only ranked as the 1st colocalized cell type for SMGs, and IgA plasma was ranked the 3rd for SMGs'

**Figure 2. Hist2Cell can map fine-grained human lung cellular architectures**
(A) Histogram depicting the average Pearson's R values for cell abundance prediction in the leave-one-donor-out cross-validation experiment conducted on the healthy human lung dataset. Hist2Cell demonstrated more accurate predictions of fine-grained cell-type abundances across locations compared to other ST prediction baselines. Error bars represent standard error across the cross-validation folds.
(B) 2D histogram plots showcasing the concordance of cell abundance between ground truth (x axis) and Hist2Cell's prediction (y axis) across all testing spots in the healthy human lung slide. Color denotes 2D histogram counts. Pearson's R denotes Pearson's correlation coefficient, and JSD denotes Jensen-Shannon divergence. Color intensity represents spot frequency. Hist2Cell showed significantly high positive correlations with several key cell types.
(C) Scatterplot for one test slide illustrating the relationship between spatial autocorrelation (Moran's *I*; x axis) and Hist2Cell's prediction performance (y axis) in the healthy human lung dataset. Color intensity represents cell abundance. Hist2Cell excelled in predicting cell types exhibiting higher spatial autocorrelation. The Pearson correlation coefficient between the x axis and the y axis is 0.74.
(D) Example H&E image for slide (ID) 9258464.
(E) Related to (D), the visualizations compare the spatial cell abundances as determined by ground truth, Hist2Cell, and DeepSpaCE* predictions for select key cell types. Color intensity represents cell abundance. Hist2Cell shows fewer false positives than DeepSpaCE*.
(F) Histogram depicting the cosine similarity and correlation between Moran's R for cell-type pairs calculated from different models' predictions and ground-truth cell abundances.
(G) Related to (F), the cluster maps are colored by the bivariate statistic Moran's R for cell-type pairs. Top: results from ground-truth cell-type abundances estimated from the spatial transcriptomics data using the Cell2location algorithm. Middle: results from predictions of Hist2Cell. Bottom: results from predictions of DeepSpaCE* (the best-performing adapted ST prediction baseline). Color intensity represents the value of the bivariate statistic Moran's R. Hist2Cell provided a more consistent cell-cell colocalization analysis with the ground-truth cell abundances.

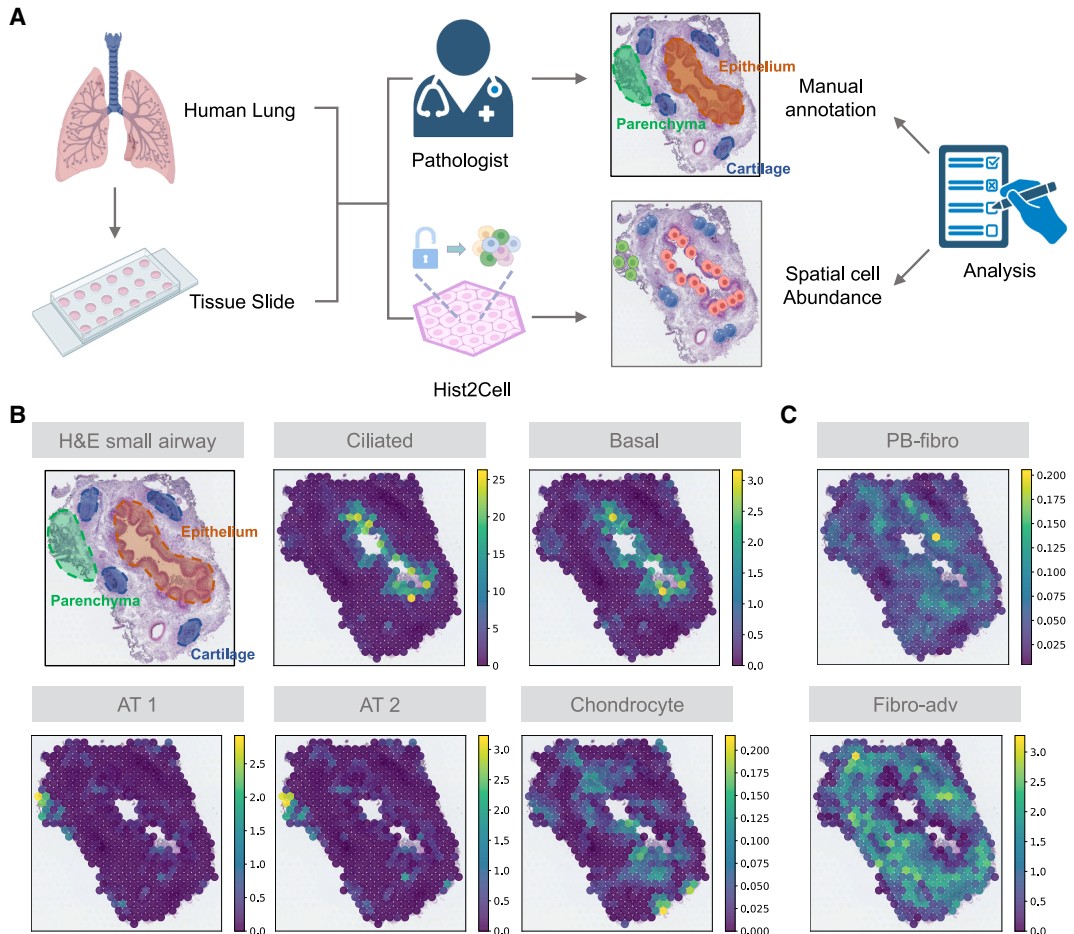

**Figure 3. Analysis workflow incorporating Hist2Cell and manual annotation**

(A) Analysis workflow incorporating Hist2Cell and manual annotation.

(B) Mapping with Hist2Cell on Visium ST of a bronchi section demonstrates the alignment of cell types with their expected anatomical structures. H&E staining and cell abundance predictions by Hist2Cell (denoted as density scores) for ciliated, basal epithelium, AT1, AT2, and chondrocyte cell types are overlaid on a histology image. Dotted lines demarcate the epithelium (pink), parenchyma (green), and cartilage (brown). Color intensity represents cell abundance.

(C) Cell abundance predictions by Hist2Cell indicate that PB-fibro cells localize around the airway epithelium. Color intensity represents cell abundance.

colocalized partner if using the cell-type abundance predicted by our method. These results suggest that the global colocalization patterns and the individual cell-type pairs with critical interactions can be identified by our method, using only histology images. Detailed cluster maps can be found in Figures S9–S11.

Additionally, by comparing Hist2Cell predictions with manual annotations from the original study[6] of the dataset (Figure 3A), we conclude that Hist2Cell's outputs can serve as reliable references for biological research, such as the discovery of macro- and micro-anatomical tissue compartments. Specifically, Hist2Cell's accurately predicted cell types correspond to their known locations, such as ciliated epithelial cells in the lumen of the airway surrounded by basal cells and alveolar types 1 and 2 (AT1 and AT2) cells in lung parenchyma, as shown in Figure 3B. Furthermore, Hist2Cell accurately localized one fibroblast population, enriched in the airways, to its specific region around the airway epithelium (peribronchial fibroblasts [PB-fibros]), as depicted in Figure 3C. This finding also validates the

clinical value of Hist2Cell, as PB-fibros are recognized as a key cell type in lung disease.[6] In Figure S12, we also compare the predictions of DeepSpaCE* with the manual annotation, which shows poor alignments between the cell types and the tissue compartments.

## Hist2Cell can generalize to external unseen breast cancer samples

To evaluate the generalizability and clinical applications of Hist2Cell, we conducted an external validation experiment using human breast cancer tissues. Specifically, the Hist2Cell model was trained on the her2st breast cancer dataset (13,620 spots in 36 sections from 8 patients)[16] to predict cell abundance for 39 fine-grained cell types and was then directly applied to the STNet breast cancer dataset (30,655 spots in 69 sections from 23 patients)[18] without any re-training. All performance comparisons, visualizations, and analyses are reported using the

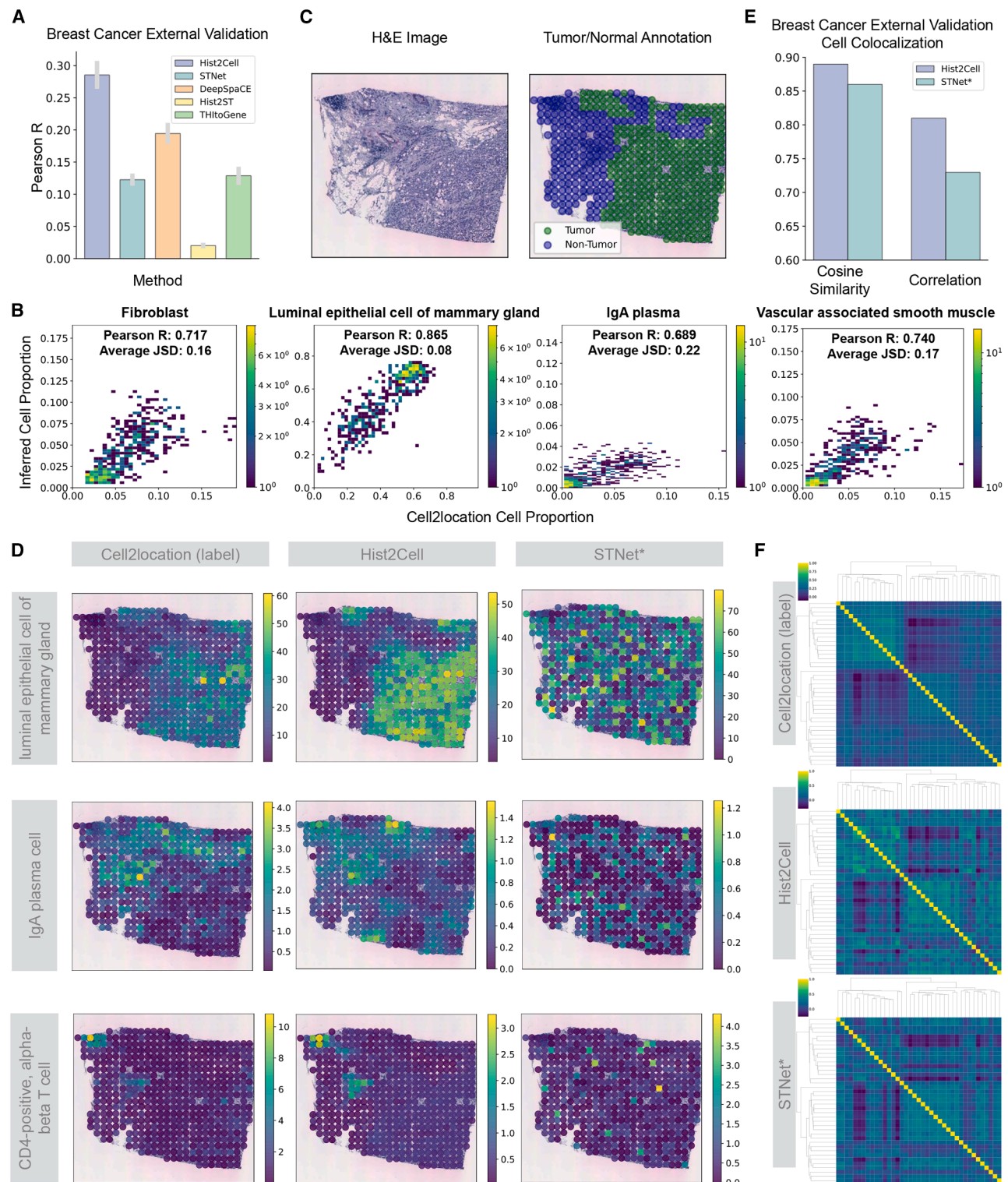

**Figure 4. Hist2Cell can generalize to external unseen breast cancer samples**

(A) A histogram representing the average Pearson's R values for cell abundance prediction in the external breast cancer dataset. Hist2Cell demonstrated superior accuracy and generalizability in predicting fine-grained cell-type abundances across different locations compared to other ST prediction baselines. Error bars represent the standard error across the cross-validation folds.

*(legend continued on next page)*

external STNet dataset. More details about the 39 cell types can be found in Figure S13.

We first assessed Hist2Cell's ability to predict fine-grained transcriptional cell types on this external dataset. The overall Pearson correlations on the external dataset revealed that Hist2Cell provides more accurate predictions of cell-type abundances across various locations compared to other methods and their variants (see Figures 4A and S14). Specifically, on the external dataset, Hist2Cell's predictions positively correlated with all 39 cell types across all samples, exhibiting an average Pearson's correlation of 0.29. On the contrary, STNet (the best-performing baseline method in Figure 4A) only has an average Pearson's correlation of 0.19 on the external dataset. Consistent with our findings on the lung dataset, a detailed per-cell-type analysis confirms that Hist2Cell outperforms baselines on a majority of cell types, underscoring the robustness of both our one-stage strategy and model architecture (Figures S15 and S16). With the top 30% cell types across all slides, the average Pearson's correlation reached 0.54 vs. 0.44 by the best counterpart, as shown in Figures S17 and S18. Moreover, despite the domain shifts across labs, we found that a group of 15 fine-grained transcriptional cell types can be predicted with high to moderate accuracy (Pearson's correlation above 0.25 for at least 50% of slides) by Hist2Cell. This suggests that Hist2Cell possesses superior generalization capabilities on external data despite technical differences in sample preparation and imaging techniques of different datasets. The favorable generalization of Hist2Cell might be attributed to its one-stage prediction strategy, model architecture, and the random subgraph sampling strategy. The one-stage prediction strategy avoided learning the compounded errors and shortcuts in the spatial gene expression and directly learned the relationship between the histological morphology features related to the fine-grained cell types to enhance the generalization of the model. The model architecture integrates both local and global correlations within tissue slides, facilitating the learning of more robust histological image representations. Additionally, the random subgraph sampling strategy preserves data diversity throughout the training process. Moreover, we need to highlight that Hist2Cell maintains significantly high positive correlations with several key cell types on the external dataset (Figure 4B), such as fibroblast as a common component in the tumor microenvironment of breast cancer[35] with a Pearson's R of 0.71 and Jensen-Shannon divergence of 0.16, luminal epithelial cells of the mammary gland that can produce basal cells upon oncogenic stress[36] with a Pearson's R of 0.87 and Jensen-Shannon

divergence of 0.08, IgA plasma cells as the major player of the tumor-immune interaction[37] with a Pearson's R of 0.69 and Jensen-Shannon divergence of 0.22, and vascular-associated smooth muscle cells with a Pearson's R of 0.74 and Jensen-Shannon divergence of 0.17. These strong correlations are consistently superior to those achieved by baseline methods, as detailed in our head-to-head comparisons in the supplemental information (Figures S19 and S20). Hist2Cell also achieves elevated accuracy in predicting spatially variable cell types (i.e., higher Moran's *I*; Figure S21) on the external dataset, corroborating findings from the human lung dataset. Beyond quantitative assessment, the ability of Hist2Cell and other methods to accurately identify locations with conspicuously high cell abundance of specific cell types was also evaluated on the external dataset. Visualizations in Figures 4C and 4D demonstrate that Hist2Cell accurately identifies the presence of cell types such as IgA plasma cells, CD4-positive helper T cells, and luminal epithelial cells of the mammary gland, with fewer false positives than other methods on the external dataset. When comparing these visualizations with manual pathology tumor/normal annotations (Figures 4C, S22, and S23), it was observed that the inferred cellular architectures closely match human expert annotations, demonstrating Hist2Cell's capability to capture intratumor heterogeneity.

Furthermore, we assess whether Hist2Cell can effectively identify colocalization among these fine-grained cell types on the external dataset. Similar to the human lung data, we used the bivariate Moran's R statistic to calculate the spatial correlation and analysis of the cluster maps on this value. Figures 4E and 4F indicate that Hist2Cell outperforms the other method, showing a global pattern more akin to that obtained from the probed ST by Cell2location and higher cosine similarities as well as correlation. As a notable example, the colocalization between effector memory CD8-positive, alpha-beta T cells with myeloid dendritic cells is a highly ranked interacting pair (Moran's R = 0.38). By directly predicting from histology, Hist2Cell also identifies the high spatial correlation between these two cell types (Moran's R = 0.26), confirming the complex cooperation among immune cells against tumor cells.[38] Specifically, with the observed cell-type abundance from ST, mature natural killer (NK) T cells were ranked as the 2nd colocalized cell type for NK cells, and they were ranked as the 1st colocalized cell type if using the cell-type abundance predicted by our method. Detailed cluster maps can be found in Figures S24–S26.

Taken together, these results suggest that the predictions from Hist2Cell uncover promising colocalization patterns

(B) 2D histogram plots showcasing the agreement between the ground-truth cell proportion (*x* axis) and that predicted by Hist2Cell (*y* axis) across all testing spots in the external breast cancer slide. Color intensity represents spot frequency.

(C and D) H&E images, manual tumor/normal annotations of slide (ID) 23508D2, and visualizations comparing the spatial cell abundances as determined by ground truth, Hist2Cell, and STNet* for certain key cell types. Hist2Cell showed fewer false positives compared to STNet*. Color intensity represents cell abundance.

(E) Histogram depicting the cosine similarity and correlation between Moran's R for cell-type pairs calculated from different models' predictions and ground-truth cell abundances.

(F) Related to (E), cluster maps are colored by the bivariate statistic Moran's R for various cell-type pairs. The top shows results from ground-truth cell-type abundances estimated from spatial transcriptomics data using the Cell2location algorithm. The middle displays results from predictions by Hist2Cell. The bottom presents results from predictions by STNet* (the best-performing adapted ST prediction baseline). Color intensity represents the value of the bivariate statistic Moran's R. Hist2Cell provided more consistent cell colocalization patterns compared to the ground-truth cell abundances.

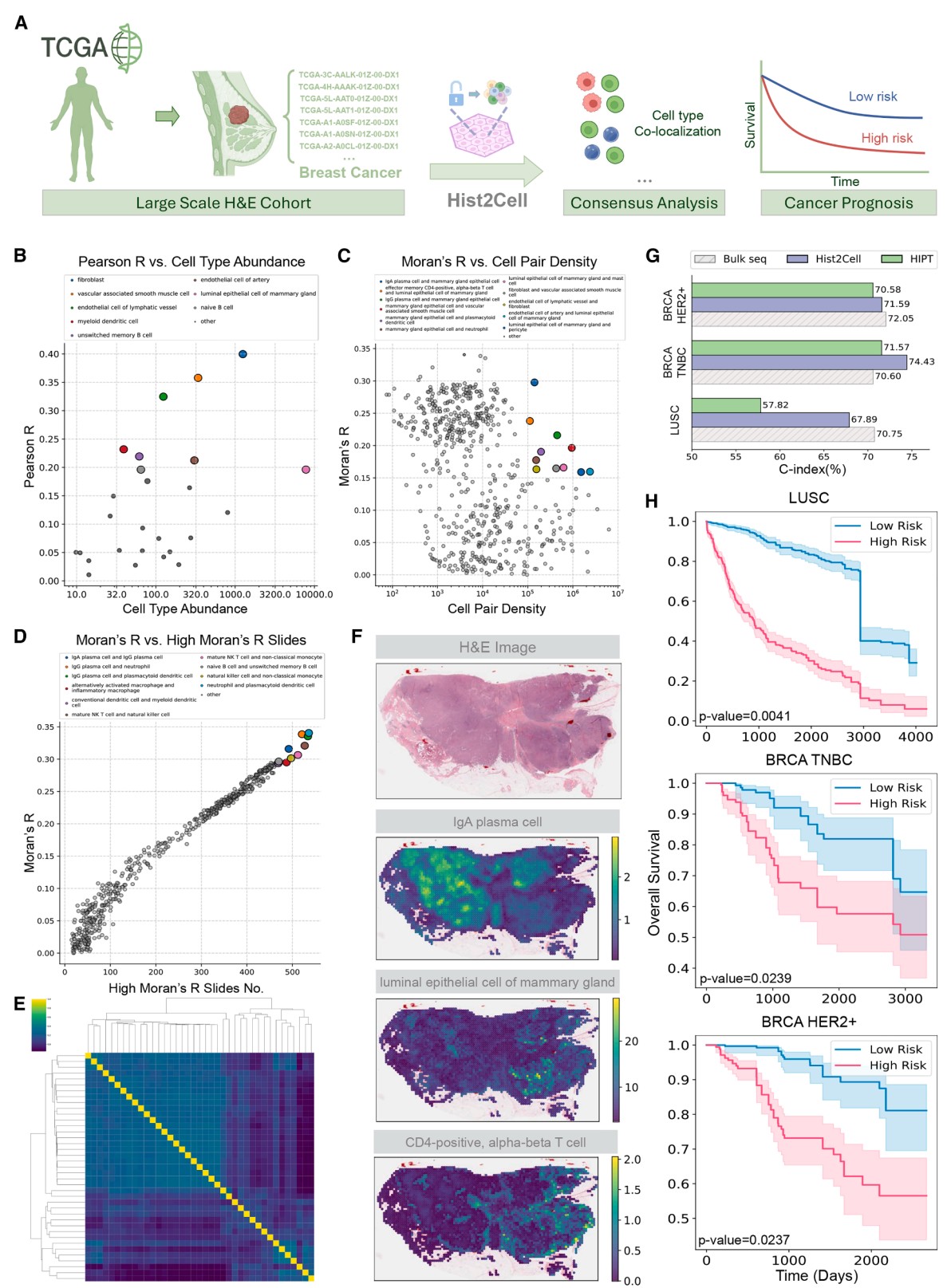

despite batch effects and technique differences, potentially facilitating the discovery of context-specific cellular cooperation and signaling on external datasets, which is crucial for its research and clinical usage.

To validate that Hist2Cell is robust to different tissue types, we expand our validation to another, entirely different tissue type—human skin—using a publicly available dataset of a non-communicable inflammatory skin disease.[39] This dataset contains 35,008 ST spots from 12 patients. Specifically, Hist2Cell was trained on 7 patients to predict cell abundance for 25 fine-grained cell types and was directly applied to the remaining 5. The results, presented in Figure S27, confirm the superior performance of our framework. Hist2Cell achieved a high average Pearson's R of 0.67, significantly outperforming all baselines. This strong performance extends to key biologically informative cell types, such as NK and cytotoxic T cells, where our model obtained correlations exceeding 0.86—a substantial improvement over the best baseline. This quantitative success is mirrored by visual inspection of the spatial maps, which clearly show that Hist2Cell more accurately captures cellular organization with noticeably fewer false positives.

For a "broader sense" of generalizability, an interesting question is whether Hist2Cell can support training in one type of cancer and prediction in another type. Actually, this setting can be referred to as pan-cancer or pan-tissue cell typing, which requires both establishing systematic cell-type nomenclature across datasets and training in multiple tissues/cancers. Two experiments have been conducted to illustrate that Hist2Cell has the potential to transfer the knowledge learned on one organ to another (from breast cancer tissue to human lung tissue in our study) in a low-data regime, a common research scenario where sufficient labeled data for a new target tissue are not available, laying the groundwork for future research that could explore pan-tissue generalization. First, we demonstrated that transferring model parameters pre-trained on breast cancer tissue to lung tissue improves performance under low-data regimes, achieving an 11% increase in the average Pearson R over training from scratch, as shown in Figure S28. Second, by merging some fine-grained cell types from both datasets into coarse-

grained categories such as epithelial cells, we applied our model trained on breast cancer directly to lung tissue. This approach yielded a PCC of 0.26 for epithelial cells, suggesting the feasibility of cross-tissue generalization with well-aligned cell-type annotations.

## Hist2Cell enables consensus cellular analysis from large-scale H&E cohorts

As Hist2Cell generalizes well on external breast cancer samples, we incorporate large-scale breast cancer histology slide datasets from TCGA repository and adopt Hist2Cell to provide consensus cellular analysis, as shown in Figure 5A. It is worth noting that there were several notable differences between TCGA and training breast cancer data. TCGA slides are collected and scanned at different institutions and only have bulk RNA-seq data that are not conducted on the same tissues used for imaging, which poses a substantial challenge for fine-grained cellular analysis. We applied Hist2Cell directly to H&E slides of 565 TCGA breast tumor samples from 565 patients without any model re-training.

To quantitatively evaluate the predictions of Hist2Cell, following STNet,[18] we first estimate the bulk cell abundance of each sample according to their bulk RNA-seq with Cell2location. The predictions from Hist2Cell were averaged into a pseudo-bulk cell abundance profile for each sample. For the cell-type abundance matrix of 39 cell types by 565 TCGA slides, we calculated Pearson's correlation between the predictions and the pseudo-bulk cell abundance profile row-wise, i.e., for each cell type across 565 slides, resulting in 39 scores. For all samples in the large cohort, the predicted pseudo-bulk cell abundance profile correlated positively with 26 of 39 cell types. The correlation was significantly positive for 13 cell types (false discovery rate [FDR] lower than 0.05). Although the Pearson R values are lower than those in the previous section, we assume this is due to the fact that bulk RNA-seq data are not generated from the same tissue as Hist2Cell's input slide. However, we believe these positive Pearson R values likely reflect Hist2Cell's ability to extract meaningful molecular information. Indeed, for a negative control, we performed shuffling of labels between slides

**Figure 5. Hist2Cell can provide consensus analysis and precise cancer prognosis on large-scale H&E TCGA cohorts**

(A) An illustration of the large-scale study process using Hist2Cell on TCGA breast cancer cohorts, demonstrating how precise consensus cellular analysis can be achieved and how cancer prognosis (survival risk prediction) can be done on slides spanning hundreds of patients.

(B) A scatterplot illustrating the relationship between cell abundance (x axis) and Hist2Cell's prediction performance (y axis) across all cases in TCGA breast cancer cohort. Hist2Cell excelled in predicting cell types with higher abundances.

(C) A scatterplot showing the relationship between cell pair density (x axis) and the average bivariate statistic Moran's R as calculated by Hist2Cell (y axis) among all cases in TCGA breast cancer cohort. Hist2Cell effectively distinguished cell-pair colocalization when the density of the cell pairs was high.

(D) A scatterplot illustrating the relationship between the number of slides with high Moran's R values (x axis, Moran's R greater than 0.20) and the average bivariate statistic Moran's R as calculated by Hist2Cell (y axis) for all cases in TCGA breast cancer cohort. Highlighted points represent colocalization patterns that Hist2Cell identified among most slides of the entire cohort.

(E) Consensus cluster maps colored by the average bivariate statistic Moran's R, calculated from Hist2Cell's predictions for cell-type pairs across all cases in TCGA breast cancer cohort. Color intensity represents the value of the average bivariate statistic Moran's R.

(F) H&E images and visualizations of spatial cell abundance predictions by Hist2Cell for a slide in TCGA breast cancer cohort. Color intensity represents cell abundance. Hist2Cell's inferred cell abundances effectively differentiated between normal and tumor regions in breast cancer tissue.

(G) Histogram depicting the average C-indices of Cox regression models predicting survival of three different cancer subtypes, LUSC, BRCA-TNBC, and BRCA-HER2+, in a 10-fold cross-validation experiment. Hist2Cell demonstrates superior performance to the previous state-of-the-art model and serves as a promising alternative to bulk RNA-seq.

(H) Cross-validated KM curves for patients split into high- and low-risk groups by the predicted risk scores of Cox regression models using predicted cell abundances from Hist2Cell. The survival risk of the low-risk group is significantly lower than that of the high-risk group ($p < 0.05$, t test).

(shuffling columns of the label matrix), and we found that this null model gives Pearson's R values all near zero (with FDR consistently failing to pass the 0.05 threshold), proving that the performance of Hist2Cell is non-trivial. Among the predicted fine-grained cell types, fibroblast cells have the highest positive correlation, as shown in Figure S29, with an average Jensen-Shannon divergence of 0.26 among all 565 TCGA breast cancer slides. We also noticed that Hist2Cell excelled in predicting cell types with higher bulk cell abundance (Figure 5B), which were generally more significant in tumor analysis. For instance, in Figure 5B, Hist2Cell shows better predictions for cell types such as fibroblasts, luminal epithelial cells of the mammary gland, and naive B cells, which is consistent with the findings in the previous section.

More importantly, by aggregating the cellular architectures from hundreds of patients, Hist2Cell unleashes the remarkable potential of large-scale public H&E cohorts and is capable of providing consensus and robust analysis for cell-cell communication analysis for biological research. As shown in Figure 5C, Hist2Cell identified cell pairs with both a high average Moran's R statistic and high cell pair density (product of bulk cell abundances) among all 565 breast cancer samples. We further investigated recurrence patterns, as depicted in Figure 5D, by visualizing cell-type colocalization relationships characterized by both high average Moran's R values and a significant number of slides with a high Moran's R (greater than 0.20). Specifically, we highlighted those relationships where the number of slides with a high Moran's R exceeded 470. Also, we show the detailed cell-type colocalization analysis provided by Hist2Cell in Figure 5E (details in Figure S30). Among them, the colocalization of T cells with myeloid cells can still be observed in the consensus setting (Moran's R with T cell ranked 3rd for myeloid cell), which is consistent with the findings in a previous study.[16] In addition, Hist2Cell inferred that cell abundances had the power to distinguish normal/tumor regions of breast cancer tissues from TCGA, as shown in Figure 5F. Moreover, we explore the impact of large-scale studies on the analytical power of Hist2Cell. As depicted in Figure S31, we visualized the cell-type colocalization analysis performed by Hist2Cell across various sample sizes. Specifically, n denotes the number of randomly sampled slides from TCGA breast cancer cohort. Using n values of 1,2,4, …,128 and the entire cohort, we generated consensus cluster maps based on Hist2Cell's predictions. It was observed that the patterns of cell-type colocalization gradually stabilized as the number of slides increased, indicating that more reliable analysis can be achieved with larger sample sizes. These analyses suggest that Hist2Cell can generalize to large-scale public H&E cohorts despite the technical differences in the samples and imaging techniques and thus provide consensus cellular analysis for biologists on the diverse large cohort studies that current ST technology cannot scale due to high cost.

### Hist2Cell offers a precise cancer prognosis from cheap clinical slides

Apart from analyzing large-scale H&E cohorts for consensus-based biological analysis, we validate that Hist2Cell is also capable of facilitating real-world applications, such as cancer prognosis, among clinical slides (Figure 5A). As previous studies

have uncovered the benefit of introducing molecular information into survival risk prediction,[40] we envision Hist2Cell's effectiveness in this significant real-world application. Specifically, we evaluate Hist2Cell's performance on survival risk prediction tasks for three important cancer subtypes, including lung squamous cell carcinoma (LUSC), triple-negative breast cancer (BRCA-TNBC), and HER2-positive breast cancer (BRCA-HER2+), with the clinical slides, bulk RNA-seq, and survival information obtained from TCGA. For the evaluation metric, we refer to the concordance index (C-index) to measure the predictive performance of correctly ranking the predicted patient risk scores with respect to overall survival.

We first show that Hist2Cell can support precise cancer survival analysis. To enhance the capability of Hist2Cell on cancer tissues, we first refer to the largest ST dataset for LUSC and breast cancer (BRCA) from the HEST-1k[41] study. Then, we applied the Cell2location[12] algorithm to these cancer ST data with scRNA-seq references from the Multi-omics Spatial Atlas of the Human Lung and the Human Breast Cell Atlas (HBCA) to obtain transcriptional fine-grained cell-type abundances, which we used as supervision for Hist2Cell on these three cancer subtypes. Note that for each cancer, we train independent Hist2Cell models. We then utilized the trained Hist2Cell model to infer the transcriptional fine-grained cell abundances from TCGA slides. To validate Hist2Cell's usefulness, we compared it to HIPT,[42] which has shown state-of-the-art survival prediction performance. Following the classic multi-instance learning paradigm for WSI analysis, we used HIPT to extract patch-level morphological features from TCGA slides. Then, the Hist2Cell-predicted cell abundance and the HIPT-extracted features were aggregated as a slide-level cell abundance profile and slide-level morphological feature, respectively. Slide-level Cox regression models were trained with the slide-level cell abundance profile and slide-level morphological feature as input. Results from 10-fold patient-level cross-validation are presented in Figure 5G. We observed that Hist2Cell outperformed HIPT among all three cancers, especially for BRCA HER2+ patients, achieving about a 10% higher C-index. Notably, we emphasize that Hist2Cell was trained on just 114 breast cancer slides and 34 lung cancer slides, far fewer than HIPT (10,000 WSIs from TCGA). This finding demonstrates that the utilization of additional ST and scRNA-seq supervisions during Hist2Cell's training could benefit precise cancer prognosis, as the predicted cell abundances reflect distinct tissue microenvironments that aid in analyzing organism function or disease processes.[6,43] Additionally, we compared with UNI,[44] a pathology foundation model pre-trained on over 100,000 diagnostic WSIs from 20 major tissue types. Though it performed better than HIPT, UNI still showed lower average performance than Hist2Cell (Table S1), with at most a 3.8% performance drop in BRCA-HER2+.

Furthermore, as the expensive bulk RNA-seq data collected from the patient's blood sample serve as an important reference in real-world cancer prognosis, we investigate if the predictions of Hist2Cell have the potential to serve as a comparable but cost-efficient alternative. In Figure 5G, we benchmarked Cox regression models trained on bulk RNA-seq data from patients, finding that Hist2Cell achieved comparable performance and even outperformed the bulk RNA-seq baseline with a 4%

C-index on BRCA-TNBC. Interestingly, for LUSC, where molecular information probably plays a significant role, both Hist2Cell and the RNA-seq model showed obvious advantages over HIPT. These results motivate us that Hist2Cell indeed has the potential to serve as a powerful yet cheaper alternative to bulk RNA-seq in cancer prognosis.

We further investigated Hist2Cell's ability to stratify patients into different risk groups. We visualized the Kaplan-Meier (KM) curve for the three cancer subtypes across the test patients, separated by the predicted risk score of the Cox regression model trained on Hist2Cell's predicted cell abundances (Figure 5H). We observed a significant difference ($p < 0.05$, $t$ test) in survival risk scores between the low-risk group and the high-risk group, and the survival curves stratified by the risk score of the Hist2Cell-based Cox regression model also show a clear separation between the low-risk and high-risk groups, suggesting that Hist2Cell captures important molecular features that relate to tissue microenvironments predictive of patient mortality and could help patient risk stratification in hospital.

We next sought to interpret the predictions of Hist2Cell in the context of survival analysis and extract biological insights regarding the relationship between fine-grained transcriptional cell populations and patient survival. We can leverage well-established deep learning interpretation methods to understand what the Hist2Cell-prediction-based model pays attention to in the fine-grained transcriptional cell populations when making predictions during the survival analysis. In Figures S32 and S33, we utilized the integrated-gradients method[45] to identify the cell types contributing to the prediction of patients' death risk in different time intervals. Specifically, we calculated the integrated gradients on each cross-validated test slide to obtain an average integrated gradient for each fine-grained transcriptional cell type. We considered these feature attributions to provide a more interpretable relationship between different cell types and the short-time/long-time survival of patients with cancer. Taking two breast cancers (Figure S32) as an example, we found out that (1) in the 10 major cell types identified by the original study of the scRNA-seq reference,[46] luminal hormone-responsive (LummHR-SCGB and LummHR-active), luminal secretory (Lumsec-KIT), Fibroblast (Fibro-matrix, Fibro-prematrix and Fibro-SFRP4), and immune (CD8-activated) cells play important roles (on the top of the ranked heatmap) in predicting patients' mortality. On the other hand, other cell types, such as vascular (vas-venous) and rare cells, showed less of a relation (in the lower part of the ranked heatmap) to the patient's survival status, which aligns with existing studies analyzing breast cancer survival.[47–52] (2) The effect of a certain cell type might vary between short-time and long-time survival analysis; for instance, we note that CD8-activated cells will have a stronger effect for long-time survival (4 times higher average integrated gradients for later survival intervals) for HER2+ cancers. Such observation provides potential biological insights for follow-up hypotheses and validation in future cancer research.

## Hist2Cell can provide super-resolved, fine-grained cellular maps

Hist2Cell, leveraging its scalability enabled by subgraph sampling, excels in inferring super-resolved cell abundance maps.

The scientific value of achieving this higher resolution lies in its ability to resolve finer structural information that is fundamentally lost in lower-resolution, binned data, as shown in the ciliated cell example in the human lung slide (Figure S34). The importance of accurately mapping these fine-scale cellular patterns for understanding complex tissue microenvironments has also been highlighted by recent work.[53] Note that, in our study, producing a higher-resolution cellular map is equivalent to predicting the relative cell abundances for more positions on the slide. Under this objective, our approach can produce high-resolution cellular maps since we can predict the relative cell abundances for a certain position based on its morphological features from the $224 \times 224$ image patches centered on this position and its neighboring image patches. As illustrated in Figure 6A, by sampling subgraphs from higher-resolution spot coordinates, Hist2Cell is capable of providing super-resolved cellular maps either by imputing (fine-tuning) lower-resolution cellular maps or by directly predicting from histology images with its pre-trained parameters. To assess Hist2Cell's proficiency in resolving higher-resolution cell abundance maps, we inferred $2\times$ cell abundance for a sample in the human lung dataset, as shown in Figure 6B. It was observed that both imputation-based and prediction-based super-resolved cellular maps provide a more detailed mapping in relation to different tissue structures in concordance with the manual annotations in Figure 3A. The high-resolution cellular maps validate the findings in the previous study that ciliated epithelial cells are in the lumen of the airway surrounded by basal cells and that AT1 and AT2 cells are in the lung parenchyma.[6] In addition, we also produced $4\times$ high-resolution cellular maps in Figure S35, and we noticed that as the resolution goes higher, we could have more detailed cellular localized patterns. Moreover, we show more detailed $8\times$ and $16\times$ cellular maps of AT1, B naive, and ciliated cells in Figures S36–S41, which further substantiates Hist2Cell's capability of depicting the super-resolved intensity of the cell population distribution within the tissue. We also provide a zoom-in inspection of the $16\times$ cellular maps of ciliated cells in Figure S42, in which we could observe that the grid distance between neighboring spots is close and even smaller than a single cell in the slide, and this resolution is almost single-cell level. In Figure S43, we further show the super-resolved cellular maps of human breast cancer tissue, which also demonstrate a more fine-grained boundary consistent with the tumor/normal annotations in Figure 4C. We observe that imputation-based maps are more consistent with the lower-resolution cellular maps, while prediction-based maps exhibit more concentrated cellular clustering patterns based on the morphological features of the histology images. Although some prior studies have focused on super-resolving low-resolution ST data,[19,21] we highlight that Hist2Cell is uniquely capable of directly providing super-resolved cellular maps from histology images without the need for ST data from the same or consecutive tissue sections of a specific patient. Specifically, Hist2Cell can offer super-resolved cellular maps even when low-resolution spatial cell abundances are unavailable, by utilizing the knowledge embedded in its pre-trained parameters. This unique capability renders super-resolved cellular mapping applicable to the vast array of H&E images routinely utilized in clinical settings. In summary, Hist2Cell provides a cost-effective method

CellPress

Cell Genomics
Technology

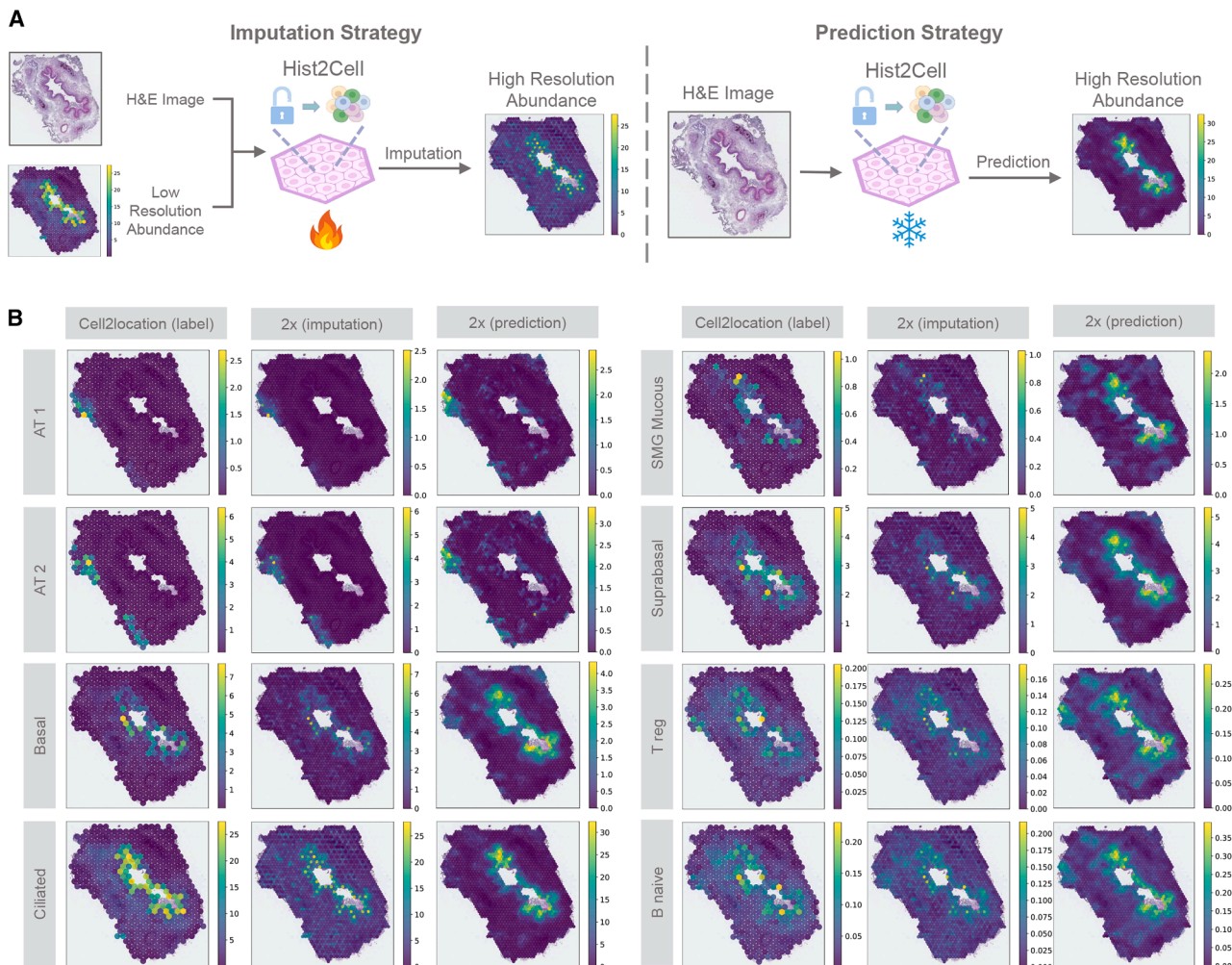

**Figure 6. Hist2Cell can provide super-resolved, fine-grained cellular maps in a neighbor-aware manner**

(A) An illustration of the two super-resolution strategies based on Hist2Cell. Left: Hist2Cell can impute from low-resolution cellular maps by fine-tuning its pretrained parameters. Right: Hist2Cell can also directly predict high-resolution cellular maps with its pre-trained parameters frozen.

(B) Visualizations comparing low-resolution ground-truth cell abundance with both imputed and directly predicted high-resolution results by Hist2Cell. Color intensity represents cell abundance. Both approaches provide a more detailed mapping in concordance with the manual annotation shown in Figure 3A, highlighting different tissue structures.

for achieving cell abundance super-resolution, enabling the acquisition of high-resolution cellular maps directly from histology images.

## DISCUSSION

In this work, we have presented Hist2Cell, a graph-transformer framework that accurately decodes fine-grained transcriptional cellular architectures (up to 40 cell types) from tissue morphological features with H&E-stained histology image inputs. Unlike analytical tools such as SiGra[54] and iIMPACT[55] that enhance or cluster a given ST dataset, Hist2Cell is a predictive tool that generalizes from paired training data to infer cellular architecture on new H&E-only slides, making spatial analysis possible for large-scale clinical archives. In addition, the capability of Hist2Cell is beyond previous digital pathology-based meth-

ods,[3–5] as they can only identify coarse-grained cell-type groups (usually 2–4 major cell types) and provide limited information for detailed analysis. A core contribution of our work is demonstrating how knowledge from disparate molecular modalities can be effectively transferred to the domain of routine histology. This is achieved via a two-stage process. First, we translate the "knowledge" from single-cell sequence signatures into spatially resolved annotations using the cell deconvolution method,[12] thereby creating a fine-grained ground-truth map of cellular abundances. Second, in a supervised learning step, Hist2Cell is trained to predict these ground-truth maps using only the corresponding H&E image patches as input. This process teaches the model to recognize the complex morphological features that are predictive of the underlying cellular composition, effectively transferring the integrated molecular knowledge into a predictive model that operates on readily available H&E images

alone. Hist2Cell produces accurate predictions for held-out donors in human lung tissues. Moreover, it can generalize to external datasets despite batch effects and technique differences in imaging without any re-training, as evidenced by the breast cancer datasets. We also find that Hist2Cell can capture cell abundance heterogeneity within tumors and different tissue structures. Although this paper focuses on healthy human lungs and three important cancers, the technical framework of Hist2Cell can be broadly applied to other tissue types.

In addition to predicting and identifying fine-grained cell-type abundance, Hist2Cell predictions can also be used for analyzing cell colocalization patterns. For example, it reveals cell-type colocalizations in the airway cartilage of human lungs. We also find it promising that Hist2Cell can make predictions on public large-scale H&E cohorts without any model re-training, despite substantial differences in the data. This opens the door to applying Hist2Cell to large existing cohorts of histology images and providing consensus biology research analysis, such as inferring important biomarkers, such as cell-cell communication, within important tissue sections, such as tumor areas. More importantly, we highlight that Hist2Cell could be applied for precise cancer prognosis, such as survival risk prediction. By revealing distinct tissue microenvironments, Hist2Cell has the potential to serve as a comparable yet cost-efficient alternative to bulk RNA-seq in clinical usage, and it could provide valuable patient survival risk stratification in real-world applications. Moreover, Hist2Cell could reveal the insightful relationship between different cell populations and patient mortality, helping in validating existing cancer research and biological phenomena, as well as providing new biological insights in an interpretable manner. In addition, due to the scalable nature of the designed framework, Hist2Cell can also infer higher-resolution cell abundance maps than current spatial transcriptome techniques.

For the first time, Hist2Cell proved that predicting transcriptional fine-grained cell types of biological interest from histology images is feasible and realistically more accurate than individual genes. Although other machine learning approaches can identify coarse-grained cell types (e.g., 2–4 cell types) from whole histology slides,[4,5] Hist2Cell can decode much more complex cell architectures in a fine-grained manner (up to 80 different cell types) by learning from ST as well as single-cell sequence signatures. With multiple datasets, we vividly demonstrated that direct cell-type prediction provides rich biological insights: our proof-of-concept study finds that the combination of ST, single-cell sequencing, and deep learning can provide direct fine-grained cellular architecture identification solely from histology images, allowing further study of complex cell interactions and variations (e.g., in the human lung dataset) as well as facilitating cancer prognosis by revealing patients' molecular attributes. Nevertheless, we also recommended image-based cell classification algorithms, such as HoverNet, if users prefer to focus on only a handful of coarse cell types for their analysis, given their well-established ability to provide cell segmentation masks.

Furthermore, the architectural design of Hist2Cell is inherently flexible and can benefit from advances in the broader field of computational pathology. To demonstrate this, we evaluated an enhanced version of our framework, replacing the original ResNet-18 encoder with a state-of-the-art pathology foundation model (UNIv2).[44] This integration yielded a substantial and consistent performance improvement across all tested datasets. For instance, on the human lung dataset, this enhanced model achieved a 16% increase in average correlation, with Pearson's R for key cell types, such as CD8 EM and ciliated cells, increasing by over 0.10, as shown in Figure S44. This performance gain was consistently replicated in our external breast cancer validation and skin disease validation (Figures S45 and S27). This not only highlights the potential for further performance gains as foundation models continue to evolve but also underscores that the core contribution of our work—the subgraph-based graph-transformer architecture for predicting cellular composition—provides a robust and adaptable platform for future innovations.

Regarding the fact that the cell types are to some extent continuous and many unknown/novel types are critical for phenotyping in cancer, Hist2Cell provides a significantly more comprehensive insight into cellular composition than previous works (2–4 major cell types), as it focuses on predicting fine-grained (up to 80 cell types) and reference-rich cell types, providing the best solution so far for this problem. In addition, Hist2Cell has the potential for novel cell-type detection by integrating post hoc out-of-distribution detection or uncertainty estimation techniques,[56,57] which could be used in potential future work.

Hist2Cell has a good generalization to unseen datasets, as verified on external breast cancer datasets and more diverse and non-standard TCGA datasets. Preliminary experimental results also show that Hist2Cell has the potential to transfer the knowledge learned on one organ to another (from breast cancer tissue to human lung tissue in our study). As different tissues contain an intersection of fine-grained cell types and share similar morphological features, an interesting direction for future work would be to explore the commonalities and differences among different spatial transcriptome datasets with aligned cell-type annotations and to build a generalized foundation model for pan-cancer analysis.

## Limitations

Despite its promising results, our study has several limitations. First, the performance of Hist2Cell is fundamentally dependent on the quality and accuracy of the ground-truth labels generated by the cell deconvolution algorithm (Cell2location). Any inaccuracies or biases in the deconvolution process, stemming from the choice of the scRNA-seq reference or the algorithm itself, will inevitably be learned by our model. Second, while Hist2Cell demonstrated strong generalization to external datasets of the same tissue type, its application to entirely new tissue types or distinct pathologies would likely require new, specific ST datasets for training or at least fine-tuning. A universal pan-tissue model remains a future goal. Third, the model is constrained to predicting the cell types present in the reference single-cell atlas used to train the deconvolution model. It cannot, in its current form, identify novel cell types or states not defined in the reference, although this could be addressed in future work by incorporating uncertainty estimation methods. Finally, the native resolution of our predictions is tied to the resolution of the ST data used for training. While we can generate higher-resolution

maps by applying the model to smaller image patches, the underlying biological patterns are learned at the spot level, and the model does not perform true single-cell segmentation or provide the precise locations of individual cells within a predictive region.

## RESOURCE AVAILABILITY

### Lead contact

Requests for further information and resources should be directed to and will be fulfilled by the lead contact, Lequan Yu (lqyu@hku.hk).

### Materials availability

This study did not generate new materials.

### Data and code availability

All datasets employed in this study are publicly accessible. The healthy lung dataset was downloaded from GitHub: https://5locationslung.cellgeni.sanger.ac.uk/. The her2st dataset was obtained from https://github.com/almaan/her2st. The STNet dataset was sourced from Mendeley Data: https://data.mendeley.com/datasets/29ntw7sh4r/5. TCGA dataset was acquired at GDC: https://portal.gdc.cancer.gov/. The scRNA-seq data from the HBCA was downloaded at CELLxGENE: https://cellxgene.cziscience.com/collections/4195ab4c-20bd-4cd3-8b3d-65601277e731. The HEST-1k dataset was acquired at Hugging Face: https://huggingface.co/datasets/MahmoodLab/hest. The skin tissue scRNA-seq data were downloaded at CELLxGene: https://cellxgene.cziscience.com/collections/34f12de7-c5e5-4813-a136-832677f98ac8.

The Hist2Cell source code is available in GitHub: https://github.com/Weiqin-Zhao/Hist2Cell and Zenodo: https://zenodo.org/records/17713950. We have uploaded the processed data and necessary files to GitHub. Furthermore, a comprehensive demo is available, providing users with a practical step-by-step tutorial for environment installation, model training, reproducing the main results of our study, and applying Hist2Cell to their own datasets.

Any additional information required to reanalyze the data reported in this paper is available from the lead contact upon request.

## ACKNOWLEDGMENTS

This work was partially supported by the Research Grants Council of the Hong Kong SAR, China (project nos. 27206123 and T45-401/22-N) and the Hong Kong Innovation and Technology Fund (project no. ITS/274/22).

## AUTHOR CONTRIBUTIONS

L.Y. and Y.H. conceived and supervised the study. W.Z. implemented the framework and performed all data analysis, with support from Z.L. and X.H. W.Z., L.Y., and Y.H. wrote the manuscript with inputs from all authors.

## DECLARATION OF INTERESTS

The authors declare no competing interests.

## DECLARATION OF GENERATIVE AI AND AI-ASSISTED TECHNOLOGIES IN THE WRITING PROCESS

During the preparation of this work, the authors used Gemini 2.5 Pro in order to polish the text language. After using this tool or service, the authors reviewed and edited the content as needed and take full responsibility for the content of the publication.

## STAR★METHODS

Detailed methods are provided in the online version of this paper and include the following:

- KEY RESOURCES TABLE
- METHOD DETAILS
  - Data description
  - Data preprocessing
  - Cell-cell colocalization analysis via spatial correlation
  - Survival analysis
  - Problem formulation
  - Super-resolved spatial cell abundance

## SUPPLEMENTAL INFORMATION

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

## STAR★METHODS

### KEY RESOURCES TABLE

| REAGENT or RESOURCE | SOURCE | IDENTIFIER |
|---|---|---|
| Software and algorithms | | |
| Scanpy | Wolf et al.[58] | https://github.com/scverse/scanpy |
| Pytorch | Paszke et al.[59] | https://github.com/pytorch/pytorch |
| STNet | He et al.[60] | https://github.com/bryanhe/ST-Net |
| HistoGene | Pang et al.[61] | https://github.com/maxpmx/HisToGene |
| THItoGene | Jia et al.[62] | https://github.com/yrjia1015/THItoGene |
| DeepSpaCE | Monjo et al.[63] | https://github.com/tmonjo/DeepSpaCE |
| Hist2Cell | This manuscript | https://github.com/Weiqin-Zhao/Hist2Cell https://zenodo.org/records/17713950 |

### METHOD DETAILS

#### Data description

##### Human lung dataset

The dataset comprising healthy human lung samples was obtained and processed as described in a prior study.[6] This dataset includes 11 slices from tissue sections encompassing the human trachea, bronchi, and both upper and lower parenchyma from 4 patients, totaling 20,770 spots. Visium Spatial Transcriptomics (ST) at 10× magnification was conducted on these slices, and Hematoxylin and Eosin (H&E) images produced during the Visium protocol were captured at a 20× magnification. The study identified 80 fine-grained cell types and estimated their absolute spatial abundances in the Visium ST data using the cell2location method (version 0.1).[12]

##### Breast cancer datasets

The her2st dataset[16] comprised HER2-positive breast tumors from 8 individuals (Patients A-H). From each individual, either 3 adjacent or 6 evenly spaced sections were obtained (totaling $n = 36$ sections), which were then subjected to Spatial Transcriptomics (ST).[64] In total, this dataset encompasses 13,620 spots, each with corresponding spatial transcriptomics data. For imaging, the her2st dataset was analyzed using the Metafer VSlide system at 20× magnification. The STNet dataset[18] included samples from 23 breast cancer patients. For each patient, the dataset contained three microscope images of slides with H&E-stained tissues and their corresponding spatial transcriptomics data, comprising a total of 30,655 spots. The biopsies were collected and processed in accordance with the methods outlined in previous work.[64]

##### TCGA large-scale cohorts

For our large-scale study, we sourced slides from The Cancer Genome Atlas (TCGA) repository and applied a two-step filtering protocol to both the breast cancer (BRCA) and lung squamous cell carcinoma (LUSC) cohorts to ensure data quality and suitability for our analyses. First, we retained only those samples with a clear one-to-one correspondence between a diagnostic H&E slide, the available bulk RNA-seq data, and the necessary clinical information (specifically survival time and subtype annotations). Second, all slides underwent a manual visual inspection, and we excluded those with insufficient tissue area (fewer than 100 valid patches after segmentation) or significant quality issues, such as poor staining or blurriness. This filtering process resulted in our final study cohorts of 565 BRCA patients (346 for BRCA HER2+, 182 for BRCA-TNBC, and the rest for other subtypes) and 487 LUSC patients. For full transparency, the list of barcodes for all selected samples is provided in our public repository. In line with previous studies,[18] the bulk RNA-seq data for each patient was utilized as the representative average molecular profile for the corresponding slide.

##### HEST-1k dataset

HEST-1k[41] is a large-scale, curated resource of paired spatial transcriptomics (ST) profiles and H&E whole-slide images. It aggregates 1,229 samples from over 150 public and internal cohorts, encompassing 26 organs and numerous cancer types, with standardized metadata and alignment. For the survival prediction task, we utilized the breast cancer and lung tissue portions of HEST-1k to train the respective Hist2Cell models that were then applied to the TCGA cohorts. Specifically, the breast cancer portion contains 108 slides, and the lung tissue portion contains 21 slides. The complete slide IDs and metadata of the slides are in Tables S2 and S3.

##### Skin disease dataset

The skin disease dataset was obtained from a prior study.[39] We selected slices that included both Visium Spatial Transcriptomics (ST) data and 20× magnification H&E images. In total, the dataset comprises 35,008 ST spots from 12 patients with various skin diseases, including lichen planus (LP), atopic dermatitis (AD), and psoriasis. We use the standardized version of this dataset from HEST-1k[41] study, the complete slide IDs and metadata of the slides are in Tables S2 and S3.

## Data preprocessing
### Spatial mapping of cell types

We employed the fine-grained spatial cell abundance, as estimated from spatial transcriptomics data, as ground truth for training and evaluating Hist2Cell. To maintain consistency with the healthy human lung dataset, we utilized the publicly available Cell2location method to estimate the fine-grained spatial and bulk cell abundances for the her2st, STNet, and TCGA datasets, respectively. The scRNA-seq data from the Human Breast Cell Atlas (HBCA)[46] served as the reference for gene expression signatures of various cell types. This single-cell transcriptomics dataset profiles 714,331 cells from 126 women, uncovering abundant pericyte, endothelial, and immune cell populations, alongside a wide diversity of luminal epithelial cell states. This makes it an excellent reference for studying mammary biology and breast cancer pathology. Specifically, this dataset offers annotations for 39 distinct cell types, encompassing both breast cancer ($n$ = 144,285) and normal breast cells ($n$ = 570,046). For the skin disease dataset, we use the scRNA-seq data from a prior study,[65] which contains 152,189 cells from 23 donors across five facial skin sites. Specifically, this dataset offers annotations for 25 distinct cell types.

### H&E image cropping

For the healthy human lung, her2st, and STNet datasets, we cropped 224 × 224 image patches from the whole-slide images. These patches were centered on the spatial transcriptomics spots to predict the fine-grained cell type abundances for each spot. For the breast cancer slides from the TCGA repository, we processed the whole-slide images into patches containing substantial tissue amounts, adhering to the pipeline established in previous studies.[66] Initially, candidate 224 × 224 patches were cropped from the whole-slide images without overlap, under a ×10 magnification. Subsequently, patches predominantly consisting of background were discarded. A pixel within a patch was classified as background if its mean RGB value was less than or equal to 220 (indicating a color close to white). A patch was excluded if over 75% of its pixels were categorized as background. Each selected patch was treated as one spot in the TCGA slides. The central pixel coordinates of each spot were utilized for graph construction, visualization, and downstream analysis in our study.

### Color normalization

To integrate multiple training/testing samples from diverse sources, such as the TCGA and ST datasets, we employ structure-preserved color normalization techniques, as proposed by Vahadane et al.[67] This method effectively normalizes color variations that typically arise from different microscopes or scanners and inconsistencies in slide preparation methods. By selecting a representative slide from the ST dataset as a reference, we standardize the color profiles of all training images to this target, thereby reducing one of the most prominent sources of batch effects in histological analyses.

### Hist2Cell design and training

Hist2Cell is constructed by leveraging the advantages and addressing the limitations of previous works focused on predicting gene expression from histology images, incorporating their key findings. Current approaches can be divided into two groups: (1) global-based methods and (2) local-based methods. In global-based methods, the core concept involves using Whole Slide Images (WSIs) to simultaneously predict an expression map for all coordinates. This enables the model to consider spatial associations between spots. However, this method suffers from severe data scarcity, making models prone to overfitting. Additionally, processing the entire WSI incurs in a high computational cost. Conversely, local-based methods rely solely on the visual information available at each coordinate for predicting gene expression. Training on spot patches, local-based methods benefit from abundant data for deep learning training. Nevertheless, such methods do not take into consideration characteristics such as the vicinity of the patch or long-range interactions, resulting in sub-optimal performance. Previous studies have also discovered that images with similar features tend to exhibit similar gene expression patterns, regardless of their location within the tissue.[68,69] Building on these findings, we propose employing k-hop subgraphs as the fundamental unit of our framework and utilize a Graph-Transformer architecture to learn both local context and distant relations, thus benefiting from the advantages of local-based and global-based methods without succumbing to their respective limitations.

To be specific, the key idea of our approach is to execute a minibatch of local subgraphs sampled from a WSI instead of using an entire WSI as a single data element. This strategy conserves computational resources and mitigates the overfitting risks commonly associated with using large WSIs, which makes our framework suitable for super-resolved cellular maps inference and improves the generalization of our framework. To better model the relationships within each subgraph and among different subgraphs, we employed a Graph-Transformer architecture. The Graph Attention layers facilitate message passing between spots and their spatial neighbors, learning the local spatial context within each subgraph, while the Transformer layers calculate the attention score among any pair of spots despite the physical distance, modeling long-range correlations existing among different subgraphs. In Supplement Figure S46 we validate the effectiveness of leveraging both local and global relations from the tissue, results showing that both the GNN part and the Transformer part contribute to the performance increase of Hist2Cell. Specifically, we further investigate the effectiveness of GNN over typical spatial smoothing in predicting fine-grained cell types. Previous research in this domain such as[26] has demonstrated the benefit of learning associations between spatially adjacent spots when predicting spatial gene expression from histology images. We propose that similar benefits extend to predicting fine-grained cell type abundances, which are inherently correlated with spatial gene expressions. GNNs are particularly suited for this task because they do not simply perform spatial smoothing; they learn advanced feature representations by integrating adjacency information with the attributes of each node (spot). This capability allows GNNs to capture complex structures and relationships within spatial data more effectively than traditional smoothing techniques. Empirical results in Supplement Figure S47 validate our hypothesis and demonstrate the effectiveness

of GNN in understanding and modeling the complex spatial interactions that are crucial for the accurate prediction of fine-grained cell type abundance. More details of the framework are illustrated in the subsequent sections.

### Spatial graph construction

As illustrated in Figure 1B, we abstract the whole-slide image (WSI) into a spatial graph. Within this spatial graph, each node corresponds to a $224 \times 224$ image patch, centered on specific spatial spots, targeted for spatial cell abundance prediction. Utilizing the spatial coordinate information of all nodes, we calculate the distances between nodes using the Euclidean distance. For each node, we select $k$ nearest nodes based on the smallest Euclidean distances, where $k = 6$ for $10\times$ Visium data and $k = 8$ for ST data.[64] Leveraging the spatial neighbor relationships, we construct an undirected graph, denoted as $G = \{V, E\}$, to represent the corresponding whole-slide image. Here, the node set $V$ represents the image patches, and the edge set $E$ indicates the spatial neighbor relationships among these patches.

### Local subgraph sampling and embedding

The core concept involves initially sampling all the nodes required for computation. We begin by randomly sampling a minibatch of central nodes, denoted as $B$, from the spatial graph $G$. Subsequently, using a pre-defined radius $k$ for local subgraphs, we gather all necessary nodes for computation, denoted as $B^k = B \cup N_1(B) \ldots \cup N_k(B)$. We use the notation $N_i(B)$ to denote a deterministic function that specifies the set of all $i$-hop neighbor nodes of all center nodes in $B$. In this study, we use $k = 2$ for both training and inference of Hist2Cell. Together with $B^k$, we obtain the spatial neighbor relationship $E_B^k$ among all nodes in $B^k$ as the corresponding subset of edge set $E$. To embed the image patches within $B^k$, we employ a feature extractor, represented as $I(\cdot)$, to learn fine-grained spatial heterogeneity in cell type abundances within the tissue. This yields a low-dimensional embedding $I_{emb} \in \mathbb{R}^{[d_{emb}, |B^k|]}$, where $d_{emb}$ is the dimension of extracted node embedding, and $I_{emb}$ denotes all node embedding of $B^k$. In our experiments, we used a ResNet-18[70] model with pre-trained ImageNet[71] weights as its initialization. The ResNet-18 model comprises 18 convolutional layers with residual connections, followed by a fully connected layer. We remove the fully connected layer and use the feature embedding before it as the node embedding in our study. Throughout the training of Hist2Cell, the parameters of the ResNet-18 model are optimized in an end-to-end fashion.

### Short-range local message aggregation

Adhering to the principles of the Graph-Transformer architecture, Hist2Cell initially employs a Graph Attention layer to learn local representations of sampled subgraphs $B^k$. As demonstrated in prior studies,[72,73] the Graph Neural Network (GNN) component of a Graph-Transformer architecture derives node representations from neighborhood features. This neighborhood aggregation in GNN plays a crucial role in learning local and short-range correlations among graph nodes. Specifically, we implement the GATv2 layer[74] for message propagation and aggregation. The message propagation and aggregation of node $i$ in $B^k$ are defined as:

$$\boldsymbol{h}_i' = f_\theta(\boldsymbol{h}_i, \text{Aggregate}(\{\boldsymbol{h}_j | j \in N_i\})) = \sigma\left(\sum_{j \in N_i} \alpha_{ij} \cdot \boldsymbol{W} \boldsymbol{h}_j\right), \tag{Equation 1}$$

where $h_i$ and $h_j$ are the node embeddings of node $i$ and node $j$ from the previous layer, and $N_i$ denotes the set of all spatial neighbor nodes of node $i$ in $B^k$. Thus, the GATv2 layer computes a weighted average of the transformed features of neighbor nodes (followed by a nonlinearity $\sigma$) as the new representation $\boldsymbol{h}_i'$ of node $i$, using normalized attention coefficients $\alpha_{ij}$. For attention coefficient calculation, a scoring function $e : \mathbb{R}^{d_{emb}} \times \mathbb{R}^{d_{emb}} \rightarrow \mathbb{R}$ computes scores for each edge $(j, i)$, indicating the importance of neighbor node $j$'s features to node $i$:

$$e(\boldsymbol{h}_i, \boldsymbol{h}_j) = \boldsymbol{a}^\top \text{LeakyReLU} \quad (\boldsymbol{W} \cdot [\boldsymbol{h}_i \| \boldsymbol{h}_j]), \tag{Equation 2}$$

where $\boldsymbol{a} \in \mathbb{R}^{2d'}$, $\boldsymbol{W} \in \mathbb{R}^{d' \times d}$ are learned during the training process, and $\|$ denotes vector concatenation. The attention scores are normalized across all neighboring nodes in $N_i$ using the softmax function, and the attention function is defined as:

$$\alpha_{ij} = \text{softmax} \quad _j(e(\boldsymbol{h}_i, \boldsymbol{h}_j)) = \frac{\exp(e(\boldsymbol{h}_i, \boldsymbol{h}_j))}{\sum_{j' \in N_i} \exp(e(\boldsymbol{h}_i, \boldsymbol{h}_{j'}))}. \tag{Equation 3}$$

### Long-range global correlation learning

In Hist2Cell, apart from the local aggregation of neighbor nodes, we also learn the long-range correlations among all sampled nodes in $B^k$. The underlying premise is that spots exhibiting similar morphological features should have similar cell abundances, regardless of their distance within the tissue. Hist2Cell employs the Transformer[75] to learn these long-range correlations, treating each node in $B^k$ as tokens. The core component of the Transformer is its multi-head attention mechanism, denoted as:

$$\text{MultiHead} \quad (Q, K, V) = [O_1, \ldots, O_{head}] W^O, \tag{Equation 4}$$

$$O_i = \text{Attention} \quad (QW_i^Q, KW_i^K, VW_i^V) = \sigma(\boldsymbol{Q}\boldsymbol{K}^T)\boldsymbol{V}. \tag{Equation 5}$$

Within this formula, *head* represents the number of parallel attention layers, and $\sigma$ denotes an activation function. Utilizing the components mentioned above, the Transformer layer is defined as:

$$Z = \text{LayerNorm} \quad (H_{\text{L}} + \text{MultiHead} \quad (H_{\text{L}}, H_{\text{L}}, H_{\text{L}})), \tag{Equation 6}$$

$$\widehat{H}_{\text{L}} = \text{LayerNorm}(Z + \text{rFF}(Z)), \tag{Equation 7}$$

where $H_{\text{L}}$ is the features of $B^k$ fed into the $L$-th Transformer layer, rFF is a row-wise feedforward layer, LayerNorm is layer normalization,[76] and $\widehat{H}_j$ represents the transformed features of all spots in $B^k$ after considering the long-range correlations.

Multi-scale Feature Fusion and Prediction In the final stage, Hist2Cell fuses multi-scale features from the feature extractor, the Graph Attention layers, and the Transformer layers using a residual block as shown in Figure 1B, which can be denoted as:

$$X_{fuse} = w_i I_{\text{fuse}} + w_g X_{GAT} + w_t X_{Trans}, \tag{Equation 8}$$

where $X_{fuse}$, $I_{\text{fuse}}$, $X_{GAT}$, and $X_{Trans}$ are the fused features, features from the feature extractor, Graph Attention part, and Transformer part, respectively, and $w_i$, $w_g$, and $w_t$ are pre-defined weights for feature fusion. Subsequently, we apply Multi-Layer Perceptron (MLP) heads to each node in $B^k$ for spatial cell abundance prediction:

$$\widehat{Y} = \text{MLP} \quad (X_{fuse}). \tag{Equation 9}$$

The Mean Squared Error (MSE) loss is employed to optimize the parameters in Hist2Cell using gradient descent. As the training of Hist2Cell is conducted on randomly sampled subgraphs, and the loss term is calculated among all sampled nodes in $B^k$, the inference phase is slightly different for Hist2Cell. During inference, every spot in the whole-slide image is sampled as the center node of subgraphs. For more reliable cell abundance prediction, we only use the predictions from the center nodes of the subgraphs and merge them into slide-level spatial cell abundances.

### Cell-cell colocalization analysis via spatial correlation

Cell-cell colocalization analysis, leveraging spatial cell abundance data, was conducted using the spatial correlation statistic, bivariate Moran's R, as implemented in the SpatialDM toolbox.[17] Specifically, to evaluate the spatial correlation between cell type pairs, we calculated the bivariate Moran's R statistic using SpatialDM. The bivariate Moran's R statistic, originally proposed for detecting ligand-receptor pairs exhibiting significant spatial co-expression, facilitates reliable analysis of cell-cell communication in spatial transcriptomics data. This statistic is an extension of techniques previously utilized in geography.[77] Utilizing the available spatial abundance data for various cell types in our study, we directly calculated the bivariate Moran's R statistic between different pairs of cell types. Specifically, the statistic is defined as:

$$\text{Global Moran's} R = \frac{\sum_i \sum_j w_{ij}(x_i - \overline{x})(y_j - \overline{y})}{\sqrt{\sum_i (x_i - \overline{x})^2}\sqrt{\sum_i (y_i - \overline{y})^2}}, \tag{Equation 10}$$

where $x_i$ and $y_j$ denote the cell abundances of cell types $x$ and $y$ at spots $i$ and $j$, respectively. The spatial weight matrix computation employs a Radial Basis Function (RBF) kernel with element-wise normalization:

$$w_{ij}^{(0)} = \exp\left\{-\frac{d_{ij}^2}{2l^2}\right\}, w_{ij} = \frac{n}{W}w_{ij}^{(0)}, \tag{Equation 11}$$

where $d_{ij}$ represents the geographical distance between spots $i$ and $j$ (Euclidean distance on spatial coordinates), $W$ is the sum of $w_{ij}^{(0)}$, and $n$ is the total number of spots. To assess the efficacy of Hist2Cell in predicting spatial cell abundance, we utilized clustermaps colored according to the bivariate Moran's R statistic. These were derived from both the ground truth spatial cell abundances and the predictions made by Hist2Cell and STNet. The pattern similarities between the clustermaps of the ground truth and those generated by Hist2Cell underscore the effectiveness of our method.

### Survival analysis

Survival analysis plays a pivotal role in clinical scenarios, offering critical insights into outcomes such as mortality, disease progression, and recovery by examining survival data that spans time periods encompassing censored time points.[78] With the advent of deep learning, leveraging whole slide images (WSIs) and patients' molecular information for survival prediction has shown promising results in various cancer types. Features extracted from these two modalities could be utilized to train a survival model, such as Cox regression models, for predicting patient outcomes. The concordance index is usually employed to evaluate the accuracy of the survival risk prediction.

### Problem formulation

Building upon prior work,[40] we applied a deep learning-based survival prediction technique that discretizes the survival time into intervals, with each interval being represented by a distinct output neuron. This approach mitigates the requirement for large minibatch sizes, enabling the model to be trained using individual observations. Specifically, for right-censored survival data, we construct a discrete-time survival model by dividing the continuous time axis into distinct intervals: $[t_0, t_1), [t_1, t_2), [t_2, t_3), [t_3, t_4)$, defined

by the quartiles of uncensored patients' survival times (in months) in each TCGA cohort. The discrete event time for each patient, indexed by $j$, with a continuous event time $T_{j,\,\text{cont}}$, is then described as:

$$T_j = r \text{ if } T_{j,\text{cont}} \in [t_r, t_{r+1}) \text{ for } r \in \{0, 1, 2, 3\} \tag{Equation 12}$$

Given the discrete-time label for the $j^{\text{th}}$ patient as $Y_j$, and the patient's slide-level feature vector $\boldsymbol{h}_{\text{final } j}$, the network's final layer applies a sigmoid activation function, modeling the hazard function as:

$$f_{\text{hazard}}\left(r | \boldsymbol{h}_{\text{final } j}\right) = P\left(T_j = r | T_j \geq r, \boldsymbol{h}_{\text{final } j}\right) \tag{Equation 13}$$

which relates to the survival function by:

$$f_{\text{surv}}\left(r | \boldsymbol{h}_{\text{final } j}\right) = P\left(T_j > r | \boldsymbol{h}_{\text{final } j}\right)$$
$$= \prod_{u=1}^{r} \left(1 - f_{\text{hazard}}\left(u | \boldsymbol{h}_{\text{final } j}\right)\right) \tag{Equation 14}$$

The model parameters are updated during training via the log likelihood function of a discrete survival model,[79] considering the censorship status of each patient ($c_j = 1$ if the patient survives beyond the follow-up period, and $c_j = 0$ if the patient dies within the recorded event time $T_j$):

$$L = -l = -c_j \cdot \log\left(f_{\text{surv}}\left(Y_j | \boldsymbol{h}_{\text{final } j}\right)\right)$$
$$- \left(1 - c_j\right) \cdot \log\left(f_{\text{surv}}\left(Y_j - 1 | \boldsymbol{h}_{\text{final } j}\right)\right)$$
$$- \left(1 - c_j\right) \cdot \log\left(f_{\text{hazard}}\left(Y_j | \boldsymbol{h}_{\text{final } j}\right)\right) \tag{Equation 15}$$

In training, we further enhance the model by up-weighting uncensored patient cases using a weighted sum of $L$ and $L_{\text{uncensored}}$:

$$L_{\text{surv}} = (1 - \beta) \cdot L + \beta \cdot L_{\text{uncensored}} \tag{Equation 16}$$

The second term in the loss function, specific to uncensored patients, is computed as:

$$L_{\text{uncensored}} = -\left(1 - c_j\right) \cdot \log\left(f_{\text{surv}}\left(Y_j - 1 | \boldsymbol{h}_{\text{final } j}\right)\right)$$
$$- \left(1 - c_j\right) \cdot \log\left(f_{\text{hazard}}\left(Y_j | \boldsymbol{h}_{\text{final } j}\right)\right) \tag{Equation 17}$$

Specifically, for the patient's slide-level feature vector $\boldsymbol{h}_{\text{final } j}$, we have slide-level transcriptional fine-grained cell abundances, slide-level morphological features, and patient-wise bulk RNA-seq in our study. To obtain the slide-level transcriptional fine-grained cell abundances and morphological features, we applied average and attention-based pooling among Hist2Cell's patch-level predictions and patch-level features extracted by HIPT[42] or UNI.[44] For the patient-wise bulk RNA-seq, we select the top 1000 highly variable expressions from the patients' whole profile.

### Concordance index

The concordance index (C-index) is a widely used metric to evaluate the predictive accuracy of survival models, particularly in scenarios involving censored data. It measures the ability of a model to correctly rank the survival times of individuals, providing an estimate of the discriminative power of the model. Formally, the C-index can be defined as the proportion of all possible pairs of individuals whose predicted survival times are correctly ordered, given that one individual has a longer observed survival time than the other. For a pair of individuals ($i,j$) with true survival times $T_i$ and $T_j$, and corresponding predicted risks $\widehat{T}_i$ and $\widehat{T}_j$, the C-index is computed as:

$$C = \frac{\sum_{i,j} \mathbf{1}\left(\widehat{T}_i < \widehat{T}_j\right) \mathbf{1}\left(T_i < T_j\right)}{\sum_{i,j} \mathbf{1}\left(T_i \neq T_j\right)}, \tag{Equation 18}$$

where $\mathbf{1}(\cdot)$ is the indicator function that returns 1 if the condition is true and 0 otherwise. A C-index value of 0.5 indicates a model with no predictive power, equivalent to random guessing, while a value of 1.0 indicates perfect concordance between predicted and actual outcomes. In the presence of censored data, the C-index is adjusted by considering only those pairs where one individual is known to have a longer survival time than the other.

### Super-resolved spatial cell abundance
#### Problem formulation

In our study, to produce higher-resolution cellular map is equivalent to predict the relative cell abundances for more positions on the slide. For instance, for the 2× resolution results in Figure 6, we predict the relative cell abundances for sub-spots, whose number is four times the original spots in the slide. Under this objective, our approach is able to produce high-resolution cellular maps as we could predict the relative cell abundances for a certain position based on its morphological features from the 224 × 224 image patches centered on this position and its neighboring image patches. For instance, for the 2× resolution results in Figure 6, for the original resolution, we crop 224 × 224 spot patches using a radius of about 112-pixel distances on the tissue, then, for producing the 2× resolution results, we will crop 224 × 224 sub-spot patches using a radius of about 56-pixel distances. We want to point out

**CelPress**

**Cell Genomics**
Technology

that this sliding-window strategy is reasonable due to two reasons: (1) Hist2Cell is trained with 224 × 224 image patches and it is likely to work as we did not change the input image size when producing high-resolution cellular maps; (2) producing high-resolution from 224 × 224 would not affect the reliability and the biological utility of the predicted fine-grained cell abundance as it is actually a relative value that is highly related to the gene expression in a specific location.

### Impute from low-resolution cellular map

As shown in Figure 6A left, given a low-resolution cellular map as priors, Hist2Cell can be fine-tuned on this map to make predictions on a super-resolved spatial graph derived from the same whole-slide image. This approach yields higher-resolution spatial cell abundances that are consistent with the low-resolution priors.

### Predict from H&E image

Furthermore, as shown in Figure 6A right, Hist2Cell can directly predict super-resolved spatial cell abundances from histology images of human lung and breast tissues, using its parameters optimized on these two tissue sections. These predictions could serve as reliable references, given Hist2Cell's strong capabilities to overcome batch effects on the external data.

### Model and computation advantage

Particularly, we also noticed that Hist2Cell shares some principles with HisToGene,[23] as both methods involve dense sliding window sampling. However, we note two major differences including: (1) The model architectures are fundamentally different. HisToGene relies solely on a Vision Transformer to model entire slides, whereas Hist2Cell combines a convolutional feature extractor with a Graph-Transformer model. This design enables Hist2Cell to extract morphology features for each spot, capture spatial contextual relations between neighboring spots, and understand representation similarities among distant tissue regions. With these capabilities, Hist2Cell demonstrates stronger representation power than HisToGene, making it more suitable for decoding complex transcriptional information from histology images; and (2) Hist2Cell has a GPU memory advantage. While HisToGene requires GPU memory that scales quadratically with the number of spots, Hist2Cell only needs a constant amount of GPU memory by using subgraphs as basic computational units. This is particularly important in super-resolution settings, where increased resolution results in a higher spot count per WSI. Indeed, our study provides 16× super-resolved results, whereas HisToGene only presents 2× super-resolution results in their study.

