## [Document S2. Transparent peer review records for Zhao et al. · Cell Genomics]

Hist2Cell: Deciphering Fine-grained Cellular Architectures from Histology Images

Wei Qin Zhao, Zhuo Liang, Xianjie Huang, Yuanhua Huang, Lequan Yu

Summary

Initial submission: Received : Jan 12, 2025

Scientific editor: Sara Rohban

First round of review: Number of reviewers: 3
Revision invited : Mar 19, 2025
Revision received : Jul 08, 2025

Second round of review: Number of reviewers: 3
Accepted : Dec 30, 2025

Data freely available: YES

Code freely available: YES

This transparent peer review record is not systematically proofread, type-set, or edited. Special characters, formatting, and equations may fail to render properly. Standard procedural text within the editor's letters has been deleted for the sake of brevity, but all official correspondence specific to the manuscript has been preserved.

Referees' reports, first round of review

Reviewer #1:

Hist2Cell's use of a Vision Graph-Transformer framework represents an advance in combining histology image analysis with spatial transcriptomics. This novel method addresses the challenge of fine-grained transcriptional cell type prediction, which is underexplored in current literature. While the approach is promising, I have some concerns as below, addressing these issues could enhance this study.

- 1) Despite validating on multiple datasets, the study predominantly focuses on human lung and breast cancer tissues. Expanding validation to additional tissue types would demonstrate broader applicability.
- 2) Although Hist2Cell is reported to generalize across datasets, the manuscript does not sufficiently address how batch effects or variations in imaging data impact performance.
- 3) While the manuscript mentions that the code is publicly available, detailed user instructions and examples for applying Hist2Cell to new datasets are not well-documented.
- 4) While deep learning models like Hist2Cell achieve high accuracy, they often lack interpretability. The manuscript could benefit from additional explanation of how model predictions align with known biological phenomena.
- 5) There are some other tools (PMID: 37699885; PMID: 38844966) that are able to do similar prediction, it is necessary to illustrate their differences.

Reviewer #2:

Overall, while the study addresses an important research topic and its findings are of interest, we have some questions and suggestions for further improvement:

Major points:

- 1) The pre-trained ResNet-18 model, which was trained on natural images ImageNet dataset, was used to extract image features from the patches. Recently studies have shown that pathology foundation models such as CTrans, UNI, Virchow remarkably improve model performance in this field. It would be of interest to investigate whether Hist2Cell performance could be enhanced by replacing ResNet-18 by such a foundation model - we do not wish to ask the authors to redo their analysis if they do not wish to, but they should discuss this at the very least.
- 2) For the survival prediction model, the authors state that they "refer to the largest ST dataset for Lung Squamous Cell Carcinoma (LUSC) and Breast Cancer (BRCA) from HEST-1k [40] study. Then, we applied the Cell2location [12] algorithm to these cancer ST data with scRNA-seq references from the Multi-omics Spatial Atlas of the Human Lung and the Human Breast Cell Atlas to obtain transcriptional fine-grained cell type abundances that we used as supervision for Hist2Cell on these three cancer subtypes. Note that for each cancer, we train independent Hist2Cell models. We then utilized the trained Hist2Cell model to infer the transcriptional fine-grained cell abundances from TCGA slides". It seems like the authors train the Hist2Cell model on these two new cohorts for the survival prediction

step. How many slides are in each dataset? This data is not referred to in the methods or the data availability statement. We suggest that the authors include the prediction performance results on these two new datasets as supplementary figures. We also suggest using the breast cancer data from HEST-1k as another external validation dataset in 2.2 Hist2Cell can Generalize to External Unseen Breast Cancer Samples. The authors could also show the external validation performance of the model trained on healthy lung samples on the lung cancer (LUSC) data they gathered from HEST-1k to examine how well a model trained on healthy samples generalizes to lung cancer samples.

Minor points:

- 1) The abstract needs to be revised. They wrote: "... trained on human lung and breast cancer spatial transcriptome datasets, Hist2Cell accurately predicts the abundance of each cell type across space in new patient samples...". This could make readers confused that the models were trained on human lung and breast cancer datasets together, while they actually trained two independent models, one for healthy lung, and one for breast cancer, and only the pre-trained model on breast cancer were externally evaluated.
- 2) In the discussion, they wrote: "We demonstrate that the knowledge within spatial transcriptomics and single-cell sequence signatures can be well captured by our deep learning method, and thus transferred to readily available H&E histology images to provide fine-grained transcriptional cellular maps". Could the author elaborate in more detail how the knowledge within spatial transcriptomics and single-cell sequence signatures were captured?
- 3) TCGA breast cancer has over 1,000 samples, why only 565 samples were included in this study?
- 4) For the preliminary results on transferring a model trained on breast to lung in Supplementary Figure 7, the figure implies that the breast cancer model is only trained on 25% of the data. Wouldn't it make more sense to show the results of the breast cancer model trained on the whole dataset?
- 5) Triple-Negative Breast Cancer (BRCA-TNBA) should be revised to BRCA-TNBC or simply TNBC.
- 6) Fig 2a: The unit of x and y axis for the Ciliated cell should be represented in the same format as other panels.

Reviewer #3:

Summary: The authors propose a method that predicts cell types from histology images. The proposed method is evaluated compared with baseline methods and manual annotations over multiple datasets. An application on survival prediction is included. In addition, the authors use the proposed method to enhance the resolution of cell type inference.

Comments

1. Figure 2. The authors compared the accuracy of the proposed method with the baseline methods, but only the average accuracy is given. Predictability could vary greatly between cell types. It would be helpful to summarize the accuracy metrics over all the 80 cell types for the proposed method compared with the baseline methods. Consider box plots or histograms. In addition, visualizations similar to Fig 2a for selected genes can be produced for the baseline methods to demonstrate how they perform compared to the proposed method.
2. Figure 2c&d. Are these R's computed over the same data as in Figure 2a? The values are much lower than those in Fig 2a. Please clarify
3. Figure 4. Similar to the comment about Figure 2, the rigor of the comparison with the baseline methods could be improved. In addition to variation across cell types, the testing dataset contains multiple patients with very different tissue structures, which introduces additional variation in performance that needs to be presented. Currently, only one subject is picked to demonstrate the performance.
4. Figure 6. (and associated supplementary figures). The author could provide an example where the super-resolution cell type inference could perform a task that is impossible with the original-resolution data. The current visualizations (including the zoomed-in one in the supplementary materials) are too high-level and do not demonstrate the added value of the super-resolution cell type inference.

Authors' response to the first round of review

Response to Reviewer #1

Comment: Summary Comment

Hist2Cell's use of a Vision Graph-Transformer framework represents an advance in combining histology image analysis with spatial transcriptomics. This novel method addresses the challenge of fine-grained transcriptional cell type prediction, which is underexplored in current literature. While the approach is promising, I have some concerns as below, addressing these issues could enhance this study.

Response:

We sincerely thank the reviewer for their positive evaluation and for recognizing the novelty and potential of our Hist2Cell framework. We also appreciate the constructive feedback, which has been invaluable for improving the manuscript. In response to the points raised, we have conducted new validation experiments to demonstrate broader applicability, added significant clarifications regarding our methodology, and enhanced our public repository with more detailed documentation. We are confident that these substantial revisions have fully addressed the reviewer's concerns and have significantly strengthened our study.

Comment: R1.1

Despite validating on multiple datasets, the study predominantly focuses on human lung and breast cancer tissues. Expanding validation to additional tissue types would demonstrate broader applicability.

Response:

We thank the reviewer for this constructive suggestion. We agree that demonstrating the broader applicability of Hist2Cell is important for strengthening our work. To address this, we have performed a comprehensive new set of experiments on an entirely different tissue type: **human skin**, using a publicly available dataset of a non-communicable inflammatory skin disease [1]. We believe these new results robustly demonstrate the **generalizability** of our framework.

For this new validation, we analyzed a dataset containing 35,008 ST spots from 12 patients. Consistent with our original methodology, we applied the Cell2location algorithm [2] with a human skin scRNA-seq atlas [3] to generate ground truth cell abundance labels for 25 cell types. We trained all methods on 7 patients and tested them on the remaining 5. Furthermore, inspired by R2.1's suggestion, we evaluated both our original model (Hist2Cell(ResNet)) and an enhanced version that utilizes a pathology foundation model encoder (**Hist2Cell(UNiv2)**) [4].

The results, presented in Figure R1, confirm the superior performance of our framework:

- **Superior Overall Performance:** As shown in Figure R1ab, both versions of Hist2Cell significantly outperform other baselines. Notably, leveraging pathology foundation model further enhances performance, with Hist2Cell(UNiv2) achieving an average Pearson's R of **0.67**, compared to **0.62** for Hist2Cell(ResNet).
- **Accurate Key Cell Type Identification:** Hist2Cell showed a high correlation for biologically informative cell types. For instance, as detailed in Figure R1c, our model achieved a Pearson's R of 0.869 for natural killer cells and 0.867 for cytotoxic T cells, representing a substantial improvement over the best-performing baseline.
- **Clear Visual Confirmation:** Beyond quantitative metrics, the visualizations of natural killer cells in Figure R1d clearly show that Hist2Cell more accurately captures the spatial patterns with fewer false positives.

In summary, the new experiments effectively demonstrate the broad applicability and robustness of the Hist2Cell. We have incorporated these findings into the revised manuscript by adding new analysis to Section 2.2 and updating Section 4.1 with a description of the skin disease dataset.

Comment: R1.2

Although Hist2Cell is reported to generalize across datasets, the manuscript does not sufficiently address how batch effects or variations in imaging data impact performance.

Response:

We thank the reviewer for raising this important point. We agree that robustness to batch effects and imaging variations is critical for any computational pathology model. We appreciate this opportunity to clarify how Hist2Cell is intrinsically designed to handle such variations through three key strategies:

Figure R1: a,b Histogram depicting the average Pearson's R values for cell abundance prediction for the skin disease dataset. An asterisk * denotes that we adapted the corresponding ST-prediction-based baseline to our one-stage strategy to predict the fine-grained cell abundances. c, 2D histogram plots showcasing the concordance of cell abundance between ground truth (x axis) and Hist2Cell's prediction (y-axis) across all testing spots in the skin disease slide. Color denotes 2D histogram counts, and JSD denotes Jensen–Shannon divergence. d, The H&E images and the visualisations comparing the natural killer cell abundances as determined by ground truth, Hist2Cell, and STNet* predictions.

- **Structure-Preserved Color Normalization:** As detailed in our Methods section, we employ a structure-preserving color normalization technique [5] as a standard preprocessing step. This method standardizes the color profiles of all histology images to a common reference, effectively mitigating color variations arising from different scanners or slide preparation methods, which are a primary source of batch effects in histological analysis.
- **Random Subgraph Sampling:** Our training strategy does not use the whole-slide image (WSI) as a single input. Instead, we sample numerous random subgraphs from each WSI. This creates highly diverse mini-batches, preventing the model from overfitting to slide-specific or batch-specific artifacts and thereby promoting better generalization.
- **Architectural Normalization Layers:** Our deep learning architecture incorporates normalization layers, such as Batch Normalization [6] in the encoder and Layer Normalization in the Transformer. These layers dynamically normalize feature distributions within mini-batches, which is a standard and powerful technique for reducing internal covariate shift and ensuring stable model performance across data from different sources.

Beyond these built-in mechanisms, the robustness of Hist2Cell is empirically demonstrated by its strong performance in our survival analysis on the large-scale TCGA cohorts (Section 2.4 in our manuscript). The TCGA slides are collected from numerous institutions, making them a challenging real-world benchmark with significant batch effects. The fact that Hist2Cell achieves promising survival prediction performance in this setting provides strong evidence of its resilience to the challenging H&E batch effects encountered in multi-center clinical applications.

Comment: R1.3

While the manuscript mentions that the code is publicly available, detailed user instructions and examples for applying Hist2Cell to new datasets are not well-documented.

Response:

We thank the reviewer for this valuable feedback regarding the usability of our software. We fully agree that clear and comprehensive documentation is essential for ensuring the reproducibility of our work and facilitating its adoption by the research community .

To address this, we have substantially enhanced our GitHub repository at <https://github.com/Weiqin-Zhao/Hist2Cell> . We have expanded our documentation with a comprehensive, step-by-step tutorial that now guides a user through the entire pipeline of applying Hist2Cell to a new, custom dataset. This enhanced tutorial includes:

- **Detailed Environment Setup:** Clear instructions for installing all required dependencies.
- **Data Preprocessing Guide:** A guide with executable scripts on how to format H&E images and spatial data.
- **Training and Inference:** A walkthrough of how to train Hist2Cell on a new dataset and how to use pre-trained models for inference, with explanations for key parameters.
- **A Complete Demo:** A demo script with example data that allows users to run the entire pipeline from start to finish, to verify their setup before using their own data.

We are confident that these significant enhancements have made our software package much more user-friendly and will substantially lower the barrier for other researchers to apply Hist2Cell to their own datasets. We have also updated Section 6 in the manuscript to reflect these improvements.

Comment: R1.4

While deep learning models like Hist2Cell achieve high accuracy, they often lack interpretability. The manuscript could benefit from additional explanation of how model predictions align with known biological phenomena.

Response:

We thank the reviewer for raising this critical point. We agree that for any deep learning model to be truly useful in a biological or clinical context, its predictions should be interpretable and align with established biological knowledge. We are grateful for the opportunity to highlight how our study addresses this from two key perspectives.

First, we validated that Hist2Cell's predictions are not just statistically accurate but also biologically coherent by comparing them against the **expert-validated biological phenomena** from the original study that published the human lung dataset [7]. As detailed in Figure R2 (also shown as Figure 3 in our manuscript), our model's outputs show a strong concordance with these established phenomena:

- Hist2Cell accurately places ciliated epithelial cells, basal cells, alveolar types 1 and 2 (AT1 and AT2) in their expected anatomical niches, precisely matching the manual annotations from the original lung atlas study.
- Additionally, the model correctly predicts the localization of PB-fibroblasts around the airway epithelium, a finding of high clinical relevance to lung disease as highlighted in the original study.

This direct comparison against ground truth established by domain experts demonstrates that Hist2Cell learns genuine histological patterns and produces biologically sound results.

Figure R2: **a**, Mapping with Hist2Cell on Visium ST of a bronchi section demonstrates the alignment of cell types with their expected anatomical structures. H&E staining and cell abundance predictions by Hist2Cell for ciliated, basal epithelium, AT1, AT2, and chondrocyte cell types are overlaid on a histology image. Dotted lines demarcate the epithelium (orange), parenchyma (green), and cartilage (blue). **b**, Cell abundance predictions by Hist2Cell indicate that PB-fibro cells localize around the airway epithelium.

Second, beyond this direct spatial validation, we ensured that the model's application to clinical tasks, such as cancer prognosis, is also **interpretable**. Using Integrated Gradients [8], we identified the specific cell types that contributed most to survival predictions, as shown in Figure RS1 and Figure RS2 (also shown as Supplementary Figures 32 and 33 in our manuscript). This analysis provides a clear link between the cellular microenvironment and patient outcomes. For example, in our breast cancer analysis, we found that:

- The cell types identified by our model as being most predictive of mortality—such as luminal hormone-responsive, fibroblast, and CD8-activated immune cells—are well-established in the literature as key players in breast cancer progression and survival [9, 10, 11, 12, 13, 14].
- Our model also revealed dynamic relationships, such as CD8-activated cells having a stronger predictive effect for long-term survival in HER2+ cancers, offering nuanced insights for future investigation.

To make these points clearer to our readers, we have expanded the Discussion section of our manuscript to better highlight how our model's predictions are validated by known biology and how its clinical predictions are interpretable. We are confident these clarifications demonstrate that Hist2Cell functions not as a "black box", but as a powerful tool for generating verifiable and biologically insightful hypotheses.

Comment: R1.5

There are some other tools (PMID: 37699885; PMID: 38844966) that are able to do similar prediction, it is necessary to illustrate their differences.

Response:

We thank the reviewer for highlighting these two relevant studies. We agree that positioning Hist2Cell relative to methods like SiGra (PMID: 37699885) [15] and iIMPACT (PMID: 38844966) [16] is crucial for clarifying its unique contribution.

The core distinction lies in the fundamental scientific goal: Hist2Cell is a **predictive** framework designed to generate new spatial insights from histology images alone, whereas SiGra and iIMPACT are **analytical** frameworks designed to refine existing, complete spatial transcriptomics (ST) datasets.

Specifically, both SiGra and iIMPACT operate on a single tissue sample where both imaging and molecular data are already available, with the primary goal of improving the analysis of that specific sample:

- **iIMPACT** improves spatial domain clustering by integrating cell-type information from the H&E image with the sample's gene expression profile. Its output is a more accurate set of spatial domain labels and a list of domain-specific genes for that sample.
- **SiGra** enhances or "denoises" the sparse molecular data of a single-cell ST sample by leveraging its accompanying high-content IHC images. Its output is an improved gene expression matrix for downstream analyses like cell-cell communication.

In sharp contrast, Hist2Cell addresses the challenge of applying spatial analysis where ST data is not available:

- Its primary goal is to **predict** fine-grained cellular architectures for new, unseen patient samples using only a standard H&E image, which it achieves by learning from paired H&E-ST datasets.
- This predictive capability enables novel, large-scale applications that are infeasible for analytical tools, such as performing spatial analysis on histology-only archives (e.g., TCGA) for clinical applications like survival prediction.

To make these crucial distinctions clear to our readers, we have added a new paragraph in the Discussion section of our revised manuscript that summarizes this comparison. We thank the reviewer again for this suggestion, as it has allowed us to more precisely situate the novelty and contribution of our work.

Response to Reviewer #2

Comment: Summary Comment

Overall, while the study addresses an important research topic and its findings are of interest, we have some questions and suggestions for further improvement.

Response:

We sincerely thank the reviewer for their positive feedback and constructive suggestions, which have been invaluable in enhancing our study. In response to the points raised, we have undertaken substantial revisions. Notably, following the reviewer's guidance, we integrated a state-of-the-art pathology foundation model into our framework, which resulted in a significant performance improvement. Alongside this major update, we have also conducted additional exploratory experiments and carefully revised the manuscript to improve clarity and rigor. We are confident that these comprehensive changes have fully addressed the reviewer's concerns and have significantly strengthened our paper.

Comment: R2.1

The pre-trained ResNet-18 model, which was trained on natural images ImageNet dataset, was used to extract image features from the patches. Recently studies have shown that pathology foundation models such as CTrans, UNI, Virchow remarkably improve model performance in this field. It would be of interest to investigate whether Hist2Cell performance could be enhanced by replacing ResNet-18 by such a foundation model - we do not wish to ask the authors to redo their analysis if they do not wish to, but they should discuss this at the very least.

Response:

We thank the reviewer for this excellent and constructive suggestion. Following their advice, we evaluated an enhanced version of our model, **Hist2Cell(UNiv2)**, which utilizes the state-of-the-art pathology foundation model UNiv2 [4] as its image encoder. The original model is referred to as Hist2Cell(ResNet) for clarity. The results, presented in Figure RS3 and Figure RS4, show significant improvements across our datasets:

- **Superior Overall Performance:** In the human lung dataset, Hist2Cell(UNiv2) achieved an average Pearson's correlation of 0.36 across all 80 cell types, a 16% improvement over the ResNet version, as shown in Figure RS3cd.
- **Key Cell Type Accuracy:** The improvement was particularly pronounced for key cell types in the lung dataset, with Pearson's R increasing by +0.11 (CD4 EM Effector), +0.10 (CD8 EM), +0.12 (ciliated cells), and +0.11 (gdT cells), as shown in Figure RS3a.
- **Enhanced Visualizations:** The predicted cell abundance maps from Hist2Cell(UNiv2) more accurately capture finescale spatial patterns, revealing more detailed and complete tissue structures, as exemplified by the ciliated cells in Figure RS3be.

A similar performance gain was consistently observed in the external breast cancer validation, confirming the generalizability of this enhancement. (Figure RS4). These results strongly support the reviewer's suggestion, demonstrating that integrating a pathology foundation model significantly enhances Hist2Cell's performance. This highlights the flexibility of our framework and its potential for further gains as foundation models evolve. We have added this new analysis to the Discussion section. We thank the reviewer again for this valuable guidance.

Comment: R2.2

For the survival prediction model, the authors state that they "refer to the largest ST dataset for Lung Squamous Cell Carcinoma (LUSC) and Breast Cancer (BRCA) from HEST-1k [40] study. Then, we applied the Cell2location [12] algorithm to these cancer ST data with scRNA-seq references from the Multi-omics Spatial Atlas of the Human Lung and the Human Breast Cell Atlas to obtain transcriptional fine-grained cell type abundances that we used as supervision for Hist2Cell on these three cancer subtypes. Note that for each cancer, we train independent Hist2Cell models. We then utilized the trained Hist2Cell model to infer the transcriptional fine-grained cell abundances from TCGA slides". It seems like the authors train the Hist2Cell model on these two new cohorts for the survival prediction step. How many slides are in each dataset? This data is not referred to in the methods or the data availability statement. We suggest that the authors include the prediction performance results on these two new datasets as supplementary figures. We also suggest using the breast cancer data from HEST-1k as another external validation dataset in 2.2 Hist2Cell can Generalize to External Unseen Breast Cancer Samples. The authors could also show the external validation performance of the model trained on healthy lung samples on the lung cancer (LUSC) data they gathered from HEST-1k to examine how well a model trained on healthy samples generalizes to lung cancer samples.

Response:

We sincerely thank the reviewer for these insightful and constructive suggestions, which have prompted us to perform significant new validations that have substantially strengthened our manuscript.

Clarification and Use of the HEST-1k Dataset: HEST-1k is a large-scale repository that aggregates numerous public ST datasets, including the human lung, her2st, and STNet datasets we used in our original analyses. For the survival prediction task, we utilized the breast cancer and lung tissue portions of HEST-1k to train the respective Hist2Cell models that were then applied to the TCGA cohorts. Specifically, the breast cancer portion contains 108 slides and the lung tissue portion contains 21 slides. We have now updated the Data Availability section to include the specific sources for each of these slides.

Following the reviewer's excellent advice, we have now also performed the suggested validation experiments on these datasets.

New External Validation on HEST-1k Breast Cancer Data: As suggested, we performed an additional external validation using the HEST-1k breast cancer data. We used the same experimental setup as in our original manuscript (training on the her2st dataset) and tested on the remaining breast cancer slides from HEST-1k (excluding the STNet and her2st datasets). The results, presented in Figure RS5, confirm our model's strong generalization capabilities. (1) As shown in Figure RS5c, Hist2Cell continues to significantly outperform all baseline methods; (2) Quantitatively, our model achieves an average Pearson's Correlation Coefficient (PCC) of over 0.4 across all cell types, and above 0.62 for key cell types (Figure RS5a); (3) Visualization of CD8 activated cells in Figure RS5b also shows that Hist2Cell accurately identifies spatial patterns of key cell type distributions, with fewer false positives compared to other methods. These results are highly consistent with our original findings on the STNet dataset, further strengthening the evidence for our model's generalization capabilities.

Generalization from Healthy Lung to Lung Cancer: We also performed the suggested experiment to test how well a model trained on healthy lung samples generalizes to a HEST-1k lung cancer slide. The results are presented in Figure RS6. The results show that generalizing from healthy to cancerous tissue is challenging for all tested methods, including ours. This is an expected outcome, given the significant biological and morphological heterogeneity between normal and malignant tissues. However, even in this very challenging cross-domain scenario, Hist2Cell still demonstrates superior performance compared to the baseline methods. This highlights the robustness and advantage of our model's design, which appears to capture more generalizable histological features that persist even across different tissue states.

Comment: R2.3

The abstract needs to be revised. They wrote: "... trained on human lung and breast cancer spatial transcriptome datasets, Hist2Cell accurately predicts the abundance of each cell type across space in new patient samples...". This could make readers confused that the models were trained on human lung and breast cancer datasets together, while they actually trained two independent models, one for healthy lung, and one for breast cancer, and only the pre-trained model on breast cancer were externally evaluated.

Response:

We thank the reviewer for pointing out this ambiguity. We agree that the original wording in the abstract could be misleading. To clarify, we indeed trained two separate and independent models for the lung and breast cancer datasets, respectively. To address this and ensure clarity, we have revised the sentence in the abstract. The revised abstract text as "... Specifically, using models trained independently on human lung and breast cancer spatial transcriptome datasets, Hist2Cell accurately predicts the abundance of each cell type across space in new patient samples..." We believe this revised wording accurately reflects our experimental design and resolves the potential confusion for the reader. We have updated the Abstract in the revised manuscript accordingly.

Comment: R2.4

In the discussion, they wrote: "We demonstrate that the knowledge within spatial transcriptomics and single-cell sequence signatures can be well captured by our deep learning method, and thus transferred to readily available H&E histology images to provide fine-grained transcriptional cellular maps". Could the author elaborate in more detail how the knowledge within spatial transcriptomics and single-cell sequence signatures were captured?

Response:

We thank the reviewer for this excellent question, which allows us to elaborate on the core mechanism of our framework. We agree that a more detailed explanation of how this knowledge transfer occurs is beneficial for the manuscript. The "knowledge" from spatial transcriptomics and single-cell signatures is captured and transferred to our deep learning model via a two-stage process:

Stage 1: From scRNA-seq to Spatial Annotations (Creating Ground Truth): First, the knowledge from "single-cell sequence signatures" is used to interpret and annotate the spatial transcriptomics (ST) data. We achieve this by employing an established deconvolution algorithm, cell2location [2]. This method uses the gene expression signatures from a reference scRNA-seq atlas to estimate the abundance of fine-grained cell types at each ST spot. The resulting cell abundance map serves as the spatially-resolved "ground truth" for training our model, effectively embedding the knowledge from both scRNA-seq and ST into a target variable.

Stage 2: From Spatial Annotations to the Hist2Cell Model (Supervised Learning): Next, this spatially-resolved ground truth is used to train our deep learning model, Hist2Cell. In this supervised learning process, Hist2Cell takes H&E histology image patches as input and is trained to predict the corresponding cell type abundance profiles generated in Stage 1. By optimizing its parameters to minimize the prediction error, the neural network learns to identify the complex morphological features within the H&E images that are predictive of the underlying cellular composition. This process effectively "transfers" the knowledge from the molecular data into the model, enabling it to later infer these compositions from new H&E images alone.

To make this process clearer to our readers, we have added sentences to the Discussion section elaborating on this twostage knowledge transfer.

Comment: R2.5

TCGA breast cancer has over 1,000 samples, why only 565 samples were included in this study?

Response:

We thank the reviewer for this important question regarding our data selection process. To ensure the quality and suitability of the data for the analyses presented in our manuscript, we applied a **two-step filtering protocol** to the TCGA-BRCA cohort, which resulted in the 565 samples used in our study. The specific selection criteria were as follows:

- **Paired Molecular and Clinical Data:** First, we retained only those samples for which there was a clear one-to-one correspondence between a diagnostic H&E slide, the available bulk RNA-seq data, and the necessary clinical information (specifically survival time and subtype annotations). This was essential for performing the downstream survival and subtype-specific analyses.
- **Image and Tissue Quality Control:** Next, all slides underwent a manual visual inspection. We excluded slides with insufficient tissue area (defined as yielding fewer than 100 valid patches after

segmentation) or those with significant quality issues, such as poor staining, blurriness, or other artifacts that could compromise the analysis.

For full transparency and reproducibility, the list of barcodes for all 565 selected samples, along with their corresponding labels, is provided in our publicly released dataset. We have added a description of these selection criteria to Section 4.2 of our revised manuscript to ensure this process is clear to our readers.

Comment: R2.6

For the preliminary results on transferring a model trained on breast to lung in Supplementary Figure 7, the figure implies that the breast cancer model is only trained on 25% of the data. Wouldn't it make more sense to show the results of the breast cancer model trained on the whole dataset?

Response:

We thank the reviewer for this insightful suggestion regarding the transfer learning experiment. Our rationale for the experiment presented in Supplementary Figure 7 was to demonstrate the most practical and impactful use case for this type of cross-tissue transfer: providing a performance boost in a **low-data regime**. In many research scenarios, sufficient labeled data for a new target tissue is not available, and our results show that in this situation, transfer learning provides a clear and significant benefit over training a model from scratch.

We agree that testing the transfer from a model trained on the full dataset is an interesting point, and we have now performed this experiment for completeness (Figure R3). As anticipated, in a high-data scenario where the target (lung) dataset also has sufficient training data, the benefit from transferring knowledge from a histologically and biologically distinct tissue is marginal. This is because the "transferable knowledge" is likely limited to general morphological patterns (e.g., epithelial vs. stromal regions) rather than the fine-grained, tissue-specific cellular features. A model with sufficient target data can learn these specific features more effectively on its own, diminishing the relative contribution of the transferred knowledge.

For these reasons, while we have now conducted the high-data regime result, we believe the original figure focusing on the data-scarce situation best illustrates the primary value of our model's transfer capabilities. We have revised the text in Section 2.2 to clarify this reasoning. We thank the reviewer for prompting this important clarification.

Figure R3: Histogram depicting the average Pearson's R values for cell abundance prediction for the human lung dataset under low and high data regimes (25% and 100% training data). Training from

pretrain denotes that we use the model weights trained on the breast cancer dataset as the initialization.

Comment: R2.7

Triple-Negative Breast Cancer (BRCA-TNBA) should be revised to BRCA-TNBC or simply TNBC.

Response:

We thank the reviewer for their careful reading and for catching this typo. We have corrected "BRCA-TNBA" to "BRCATNBC" throughout the entire manuscript, including in the text, figures, and tables, to ensure accuracy.

Comment: R2.8

Fig 2a: The unit of x and y axis for the Ciliated cell should be represented in the same format as other panels.

Response:

We thank the reviewer for their careful observation and for pointing out this formatting inconsistency. The reviewer is correct. While the scale for the Ciliated cell panel in Figure 2a differs due to its distinct abundance range, the axis label format should be uniform for clarity and professionalism. We have corrected this by revising the plot to display the tick labels with one decimal place (e.g., "10.0", "20.0") to match the style of the other panels. The updated Figure 2a has been included in the revised manuscript. We appreciate the reviewer's help in improving the quality of our figures.

Response to Reviewer #3**Comment: Summary Comment**

The authors propose a method that predicts cell types from histology images. The proposed method is evaluated compared with baseline methods and manual annotations over multiple datasets. An application on survival prediction is included. In addition, the authors use the proposed method to enhance the resolution of cell type inference.

Response:

We thank the reviewer for the dedicated summary of our work's key contributions, including the prediction of cell types from histology images, the comprehensive evaluations, and the applications in survival prediction and super-resolution mapping. We are grateful for the insightful feedback and have addressed each of the reviewer's suggestions in the point-by-point responses that follow.

Comment: R3.1

Figure 2. The authors compared the accuracy of the proposed method with the baseline methods, but only the average accuracy is given. Predictability could vary greatly between cell types. It would be helpful to summarize the accuracy metrics over all the 80 cell types for the proposed method compared with the baseline methods. Consider box plots or histograms. In addition, visualizations similar to Fig 2a for selected genes can be produced for the baseline methods to demonstrate how they perform compared to the proposed method.

Response:

We thank the reviewer for this excellent suggestion. We agree that a more granular, per-cell-type comparison is crucial for a rigorous evaluation, and we have performed new analyses to address this.

Per-Cell-Type Accuracy Comparison: Following the reviewer's suggestion, we have generated histograms of the Pearson's correlation coefficients across all 80 cell types for Hist2Cell and all baseline methods.

- This new analysis reveals that Hist2Cell outperforms the original baseline methods on **73 out of 80** cell types, many by a significant margin (Figure RS7).
- To dissect the source of this improvement, we also created "adapted" baselines using our one-stage prediction strategy. While this adaptation improves their performance, Hist2Cell still surpasses these stronger, "adapted" baselines on most cell types (new Figure RS8). This demonstrates that our performance gain stems from both our strategy advantage (one-stage prediction) and our **model advantage** (Graph-Transformer architecture).
- These conclusions are consistently replicated in our breast cancer external validation, with new summary histograms also provided in Figure RS9 and Figure RS10.

Visual Comparison for Key Cell Types: We have also produced the requested visualizations analogous to Figure 2a for the baseline methods. These new head-to-head scatter plots are now available in Figure RS11 and Figure RS12. This analysis shows that even when compared against the strongest "adapted" baseline, Hist2Cell achieves remarkable improvements in Pearson's R for all key cell types, including for CD4 EM Effector (**+0.16**) and gdT cells (**+0.28**). As before, these findings hold true in the external breast cancer validation (Figures RS13 and RS14).

We are confident that these new, detailed analyses provide a more rigorous and comprehensive comparison that fully supports our claims. We have updated Sections 2.1 and 2.2 in our manuscript to

refer to these new supplementary figures. We thank the reviewer again for prompting these valuable additions.

Comment: R3.2

Figure 2c&d. Are these R's computed over the same data as in Figure 2a? The values are much lower than those in Fig 2a. Please clarify

Response:

We thank the reviewer for their careful reading and for the opportunity to clarify these important points about our results.

Regarding Figure 2c (Average Performance): The Pearson's R value in Figure 2c is an average computed over the same test data, but it represents a different summary than Figure 2a. Figure 2a showcases the performance on four key cell types with strong visual features on a specific slide, whereas Figure 2c shows the average performance across all 80 finegrained transcriptional cell types and all test samples. The average is lower because a few of these 80 cell types lack distinct morphological correlates in H&E images, making them inherently more challenging to predict. To provide a more practical measure of performance, our manuscript includes a focused analysis of the top 30% most predictable cell types. In this analysis, our model achieves a much higher average Pearson's correlation of **0.50**, a 16% improvement over the 0.43 achieved by the best-performing baseline. Furthermore, the value of our predictions is confirmed by their strong alignment with expert manual annotations (Figure 3), where our inferred cellular maps accurately recapitulate known tissue structures.

Regarding Figure 2d (Performance vs. Spatial Variation): We thank the reviewer for pointing out the potential for confusion. Our original intent for Figure 2d was to select a sample that most clearly illustrated the important relationship between a cell type's spatial autocorrelation (Moran's I) and our model's prediction accuracy. However, to improve consistency and directly address the reviewer's concern, we have now revised Figure 2d to use the same patient sample as presented in Figure 2a. We are pleased to report that the conclusion remains identical: our model's performance is strongest for cell types that exhibit higher spatial variation, which are often the most biologically significant. We have updated the manuscript and figures to reflect these changes and believe these clarifications significantly improve the rigor and clarity of our results.

Comment: R3.3

Figure 4. Similar to the comment about Figure 2, the rigor of the comparison with the baseline methods could be improved. In addition to variation across cell types, the testing dataset contains multiple patients with very different tissue structures, which introduces additional variation in performance that needs to be presented. Currently, only one subject is picked to demonstrate the performance.

Response:

We thank the reviewer for this excellent suggestion. We agree that demonstrating the model's performance on multiple patients is crucial for showcasing its robustness to inter-patient and tissue structure variations. To provide this comprehensive evidence, we have included detailed visualizations for two additional, representative patient samples in Figure RS15 and Figure RS16. These new figures consistently reinforce our original findings. The visualizations show that Hist2Cell accurately localizes key, biologically-relevant cell populations, such as IgA plasma cells and CD4-positive helper T cells, while producing noticeably fewer false positives than the best-performing adapted baseline, STNet*. Crucially, a comparison with the pathologist's annotations shows that the cellular architectures inferred by Hist2Cell align closely with expert-defined tumor boundaries. This demonstrates the model's capability to reliably capture intratumor heterogeneity across different patients, not just on a single selected example. We believe these new figures in the supplement provide the rigorous, multi-patient validation that the reviewer requested. We have updated the manuscript to refer to this new supplementary evidence in Section 2.2 and thank the reviewer for prompting this important addition.

Comment: R3.4

Figure 6. (and associated supplementary figures). The author could provide an example where the super-resolution cell type inference could perform a task that is impossible with the original-resolution data. The current visualizations (including the zoomed-in one in the supplementary materials) are too high-level and do not demonstrate the added value of the super-resolution cell type inference.

Response:

We thank the reviewer for this excellent question, which prompts us to more clearly articulate the fundamental value of super-resolution inference. As illustrated in one real example of the Ciliated cell in the human lung slide (Figure R4), the primary advantage of our approach is its ability to **resolve finer structural information** that is fundamentally lost in lower resolution, binned data. At the original resolution, the prediction for a single spot is a "mixed" average value, making a key task impossible: discerning the cellular micro-architecture within the spot. From this averaged signal, one cannot distinguish whether the spot represents a sharp boundary between two cell populations or a smooth, intermixed transition zone; nor can one locate individual, sparsely dispersed cells. Our super-resolution inference addresses this challenge by predicting abundances at multiple sub-locations within a single low-resolution spot. This allows the model to "un-mix" the averaged signal and thereby reconstruct finer cellular organization patterns. As highlighted by a recent Nature paper [17], accurately mapping these fine-scale cellular patterns is critical for understanding complex tissue microenvironments and elucidating tissue organization. Therefore, the key added value of our method is advancing the analysis from a **coarse-grained compositional view** to a **fine-grained architectural view** of the tissue. We have revised Section 2.5 in our manuscript to elaborate on this capability in greater detail. We thank the reviewer again for their valuable suggestion.

Figure R4: Super-resolution reveals fine-scale cellular architecture missed by low-resolution methods. This figure shows the predicted ciliated cell abundance from our human lung super-resolution experiment. We show a real example of Ciliated cells in the human lung slide. Low-resolution predictions (bottom left) are averaged signals that cannot distinguish between different micro-architectures or locate individual, sparsely dispersed cells. In contrast, our super-resolution approach (bottom middle) reconstructs these finer structural details by predicting at multiple sub-locations, resolving this ambiguity and providing a more faithful view of the tissue's organization.

References

- [1] Alexander Schäbitz, C Hillig, M Mubarak, Manja Jargosch, Ali Farnoud, Emanuele Scala, Nils Kurzen, Anna Caroline Pilz, Nayanika Bhalla, Jenny Thomas, et al. Spatial transcriptomics landscape of lesions from non-communicable inflammatory skin diseases. *Nature Communications*, 13(1):7729, 2022.
- [2] Vitalii Kleshchevnikov, Artem Shmatko, Emma Dann, Alexander Aivazidis, Hamish W King, Tong Li, Rasa Elmentaite, Artem Lomakin, Veronika Kedlian, Adam Gayoso, et al. Cell2location maps fine-grained cell types in spatial transcriptomics. *Nature biotechnology*, 40(5):661–671, 2022.
- [3] Clarisse Ganier, Pavel Mazin, Gabriel Herrera-Oropeza, Xinyi Du-Harpur, Matthew Blakeley, Jeyrroy Gabriel, Alexander V Predeus, Batuhan Cakir, Martin Prete, Nasrat Harun, et al. Multiscale spatial mapping of cell populations across anatomical sites in healthy human skin and basal cell carcinoma. *Proceedings of the National Academy of Sciences*, 121(2):e2313326120, 2024.
- [4] Richard J Chen, Tong Ding, Ming Y Lu, Drew FK Williamson, Guillaume Jaume, Andrew H Song, Bowen Chen, Andrew Zhang, Daniel Shao, Muhammad Shaban, et al. Towards a general-purpose foundation model for computational pathology. *Nature Medicine*, 30(3):850–862, 2024.

- [5] Abhishek Vahadane, Tingying Peng, Shadi Albarqouni, Maximilian Baust, Katja Steiger, Anna Melissa Schlitter, Amit Sethi, Irene Esposito, and Nassir Navab. Structure-preserved color normalization for histological images. In *2015 IEEE 12th international symposium on biomedical imaging (ISBI)*, pages 1012–1015. IEEE, 2015.
- [6] Sergey Ioffe. Batch normalization: Accelerating deep network training by reducing internal covariate shift. *arXiv preprint arXiv:1502.03167*, 2015.
- [7] Elo Madisson, Amanda J Oliver, Vitalii Kleshchevnikov, Anna Wilbrey-Clark, Krzysztof Polanski, Nathan Richoz, Ana Ribeiro Orsi, Lira Mamanova, Liam Bolt, Rasa Elmentaite, et al. A spatially resolved atlas of the human lung characterizes a gland-associated immune niche. *Nature Genetics*, 55(1):66–77, 2023.
- [8] Mukund Sundararajan, Ankur Taly, and Qiqi Yan. Axiomatic attribution for deep networks. In *International conference on machine learning*, pages 3319–3328. PMLR, 2017.
- [9] HR Ali, E Provenzano, S-J Dawson, FM Blows, B Liu, M Shah, HM Earl, CJ Poole, Louise Hiller, Janet A Dunn, et al. Association between cd8+ t-cell infiltration and breast cancer survival in 12 439 patients. *Annals of oncology*, 25(8):1536–1543, 2014.
- [10] Marit J van Elsas, Jim Middelburg, Camilla Labrie, Jessica Roelands, Gaby Schaap, Marjolein Sluijter, Ruxandra Tonea, Vitalijs Ovcinnikovs, Katy Lloyd, Janine Schuurman, et al. Immunotherapy-activated t cells recruit and skew late-stage activated m1-like macrophages that are critical for therapeutic efficacy. *Cancer Cell*, 42(6):1032–1050, 2024.
- [11] Balaji Virassamy, Franco Caramia, Peter Savas, Sneha Sant, Jianan Wang, Susan N Christo, Ann Byrne, Kylie Clarke, Emmaline Brown, Zhi Ling Teo, et al. Intratumoral cd8+ t cells with a tissue-resident memory phenotype mediate local immunity and immune checkpoint responses in breast cancer. *Cancer Cell*, 41(3):585–601, 2023.
- [12] Rachel J Buchsbaum and Sun Young Oh. Breast cancer-associated fibroblasts: where we are and where we need to go. *Cancers*, 8(2):19, 2016.
- [13] Dakai Yang, Jing Liu, Hui Qian, and Qin Zhuang. Cancer-associated fibroblasts: from basic science to anticancer therapy. *Experimental & Molecular Medicine*, 55(7):1322–1332, 2023.
- [14] Zhaoze Guo, Han Zhang, Yiming Fu, Junjie Kuang, Bei Zhao, LanFang Zhang, Jie Lin, Shuhui Lin, Dehua Wu, and Guozhu Xie. Cancer-associated fibroblasts induce growth and radioresistance of breast cancer cells through paracrine il-6. *Cell Death Discovery*, 9(1):6, 2023.
- [15] Ziyang Tang, Zuotian Li, Tieying Hou, Tonglin Zhang, Baijian Yang, Jing Su, and Qianqian Song. Sigra: single-cell spatial elucidation through an image-augmented graph transformer. *Nature Communications*, 14(1):5618, 2023.
- [16] Xi Jiang, Shidan Wang, Lei Guo, Zhuoyu Wen, Liwei Jia, Lin Xu, Guanghua Xiao, and Qiwei Li. Integrating image and molecular profiles for spatial transcriptomics analysis. *bioRxiv*, pages 2023–06, 2023.
- [17] Katherine Benjamin, Aneesha Bhandari, Jessica D Kepple, Rui Qi, Zhouchun Shang, Yanan Xing, Yanru An, Nannan Zhang, Yong Hou, Tanya L Crockford, et al. Multiscale topology classifies cells in subcellular spatial transcriptomics. *Nature*, 630(8018):943–949, 2024.

[18] Tapsi Kumar, Kevin Nee, Runmin Wei, Siyuan He, Quy H Nguyen, Shanshan Bai, Kerrigan Blake, Yanwen Gong, Maren Pein, Emi Sei, et al. A spatially resolved single cell genomic atlas of the adult human breast. *bioRxiv*, pages 2023–04, 2023.

Response Supplementary

Figure RS1: Average integrated gradients (representing the feature attribution of the Hist2Cell-prediction-based Cox regression model) for the survival analysis of BRCA TNBC and BRCA HER2+. The Y-axis is the different fine-grained transcriptional cell types and the X-axis is the four survival intervals ranging from shortest to longest survival time among the patients. The fine-grained transcriptional cell types are ranked according to their importance (mean value) among four intervals. Interesting observations could be found: (1) in the ten 10 major cell types identified by the original study of the scRNA-seq reference [18], luminal hormone-responsive (LumHR-), luminal secretory (LumSec-), fibroblasts (fibro-) and immune (CD8activated) cells play important roles (on the top of the ranked heatmap) in predicting patients' mortality while other cell types, like the vascular (vas-venous) and the rare cells, less related (on the lower part of the ranked heatmap) to the patient's survival status, this aligns with the existing studies analyzing breast cancer survival [9, 10, 11, 12, 13, 14]; (2) the effect of a certain cell type might vary between short-time and long-time survival analysis, for instance, we note that CD8-activated cells will have a stronger effect for long-time survival (4 times bigger average integrated gradients for later survival intervals) for HER2+ cancers, such observation provides potential

biological insights for future cancer research and probably motivates customized BRCA-HER2+ treatment plan to promote the proliferation of certain cell types.

Figure RS2: Average integrated gradients (representing the feature attribution of the modelname-prediction-based Cox regression model) for the survival analysis of LUSC. The Y-axis is the different fine-grained transcriptional cell types and the X-axis is the four survival intervals ranging from shortest to longest survival time among the patients. The fine-grained transcriptional cell types are ranked according to their importance (mean value) among four intervals.

Figure RS3: Performance comparison on human lung dataset. Hist2Cell(UNIV2) is the enhanced version of our framework using the pathology foundation model as image encoder, while Hist2Cell(ResNet) uses ImageNet pretrained ResNet as the image encoder. a, 2D histogram plots showcasing the concordance of cell abundance between ground truth (x-axis) and model's prediction (y-axis) across all testing spots in the healthy human lung slide. Color denotes 2D histogram counts. Pearson's R denotes Pearson's

correlation coefficient, and JSD denotes Jensen–Shannon divergence. b, Example H&E image for slide (ID) 9258464. c,d, Histogram depicting the average Pearson’s R values for cell abundance prediction in the leave-one-donor-out cross-validation experiment conducted on the healthy human lung dataset. * means adapting the two-stage ST prediction baseline to our one-stage prediction strategy. e, Related to (b), the visualizations comparing the key spatial cell abundances as determined by ground truth, Hist2Cell, and DeepSpaCE* predictions. Hist2Cell shows fewer false positives than DeepSpaCE*(the best-performing adapted ST prediction baseline).

Figure RS4: Performance comparison on external unseen breast cancer dataset. Hist2Cell(UNiv2) is the enhanced version of our framework using the pathology foundation model as image encoder, while Hist2Cell(ResNet) uses ImageNet pretrained ResNet as the image encoder. a, 2D histogram plots showcasing the concordance of cell abundance between ground truth (x-axis) and model’s prediction (y-axis) across all testing spots in the breast cancer slide. Color denotes 2D histogram counts. Pearson’s R denotes Pearson’s correlation coefficient, and JSD denotes Jensen–Shannon divergence. b, Expert manual tumor/normal annotation for slide (ID) 23508D2. c,d, Histogram representing the average Pearson’s R values for cell abundance prediction in the external breast cancer dataset. * means adapting the two-stage ST prediction baseline to our one-stage prediction strategy. e, Related to (b), the visualizations comparing the key spatial cell abundances as determined by ground truth, Hist2Cell, and STNet* predictions. Hist2Cell shows fewer false positives than STNet* (the best-performing adapted ST prediction baseline).

Figure RS5: Performance comparison on external unseen breast cancer dataset from HEST-1k dataset. Hist2Cell(UNiv2) is the enhanced version of our framework using the pathology foundation model as image encoder, while Hist2Cell(ResNet) uses ImageNet pretrained ResNet as the image encoder. a, 2D histogram plots showcasing the concordance of cell abundance between ground truth (x-axis) and model's prediction (y-axis) across all testing spots in the breast cancer slide. Color denotes 2D histogram counts. Pearson's R denotes Pearson's correlation coefficient, and JSD denotes Jensen–Shannon divergence. b, Visualizations comparing the spatial cell abundances as determined by ground truth, Hist2Cell, and DeepSpaCE* predictions for CD8 activated cells. Hist2Cell shows fewer false positives than DeepSpaCE*(the best-performing adapted ST prediction baseline). c,d, Histogram representing the average Pearson's R values for cell abundance prediction in the external breast cancer dataset. * means adapting the two-stage ST prediction baseline to our one-stage prediction strategy.

Figure RS6: a,b Histograms depicting the average Pearson's R values for cell abundance prediction in the healthy-to-cancerous lung generalization experiment. An asterisk (*) denotes that the corresponding ST-prediction-based baseline was adapted to our one-stage strategy.

Figure RS7: Histogram representing the average Pearson's R values for all methods of all cell types on the human lung dataset.

Figure RS8: Histogram representing the average Pearson's R values for all adapted (*) baseline methods and Hist2Cell of all cell types on the human lung dataset.

Figure RS9: Histogram representing the average Pearson's R values for all methods of all cell types on the external breast cancer validation.

Figure RS10: Histogram representing the average Pearson's R values for all adapted (*) baseline methods and Hist2Cell of all cell types on the external breast cancer validation.

Figure RS11: 2D histogram plots showcasing the concordance of cell abundance between ground truth (x-axis) and prediction of Hist2Cell and baseline methods (y-axis) across all testing spots in the healthy human lung slide. Color denotes 2D histogram counts. Pearson's R denotes Pearson's correlation coefficient, and JSD denotes Jensen–Shannon divergence.

Figure RS12: 2D histogram plots showcasing the concordance of cell abundance between ground truth (x-axis) and prediction of Hist2Cell and the "adapted" (*) baseline methods (y-axis) across all testing spots in the healthy human lung slide. Color denotes 2D histogram counts. Pearson's R denotes Pearson's correlation coefficient, and JSD denotes Jensen–Shannon divergence.

Figure RS13: 2D histogram plots showcasing the concordance of cell abundance between ground truth (x-axis) and prediction of Hist2Cell and the baseline methods (y-axis) across all testing spots in the external breast cancer slide. Color denotes 2D histogram counts. Pearson's R denotes Pearson's correlation coefficient, and JSD denotes Jensen–Shannon divergence.

Figure RS14: 2D histogram plots showcasing the concordance of cell abundance between ground truth (x-axis) and prediction of Hist2Cell and the "adapted" (*) baseline methods (y-axis) across all testing spots in the external breast cancer slide. Color denotes 2D histogram counts. Pearson's R denotes Pearson's correlation coefficient, and JSD denotes Jensen-Shannon divergence.

Figure RS15: The H&E image, expert manual annotation, and the visualizations comparing the spatial key cell abundances as determined by ground truth, Hist2Cell, and STNet* predictions of slide (ID) 24044D2. Hist2Cell shows fewer false positives than STNet*(the best-performing adapted ST prediction baseline)

Figure RS16: The H&E image, expert manual annotation, and the visualizations comparing the key spatial cell abundances as determined by ground truth, Hist2Cell, and STNet* predictions of slide (ID) 23287C2. Hist2Cell shows fewer false positives than STNet*(the best-performing adapted ST prediction baseline)

Referees' report, second round of review

Reviewer #1: The authors have addressed my concerns. It would be recommended to polish and improve the writing.

Reviewer #2: Comments enter in this field will be shared with the author; your identity will remain anonymous.

Reviewer #3: The authors have satisfactorily addressed my previous comments.